# Adversarial Graph Fusion for Incomplete Multi-view Semi-supervised Learning with Tensorial Imputation

**Zhangqi Jiang**[1]  **Tingjin Luo**[1,∗]  **Xu Yang**[2]  **Xinyan Liang**[3]

[1]National University of Defense Technology  [2]Southeast University  [3]Shanxi University

sxdxjzq@gmail.com, tingjinluo@hotmail.com, xuyang_palm@seu.edu.cn
liangxinyan48@163.com

## Abstract

View missing remains a significant challenge in graph-based multi-view semi-supervised learning, hindering their real-world applications. To address this issue, traditional methods introduce a missing indicator matrix and focus on mining partial structure among existing samples in each view for label propagation (LP). However, we argue that these disregarded missing samples sometimes induce discontinuous local structures, *i.e.*, sub-clusters, breaking the fundamental smoothness assumption in LP. Consequently, such a **S**ub-**C**luster **P**roblem (SCP) would distort graph fusion and degrade classification performance. To alleviate SCP, we propose a novel incomplete multi-view semi-supervised learning method, termed **AGF-TI**. Firstly, we design an adversarial graph fusion scheme to learn a robust consensus graph against the distorted local structure through a min-max framework. By stacking all similarity matrices into a tensor, we further recover the incomplete structure from the high-order consistency information based on the low-rank tensor learning. Additionally, the anchor-based strategy is incorporated to reduce the computational complexity. An efficient alternative optimization algorithm combining a reduced gradient descent method is developed to solve the formulated objective, with theoretical convergence. Extensive experimental results on various datasets validate the superiority of our proposed **AGF-TI** as compared to state-of-the-art methods. Code is available at `https://github.com/ZhangqiJiang07/AGF_TI`.

## 1 Introduction

Multi-view data encodes complementary information from heterogeneous sources or modalities, significantly enhancing the performance of downstream tasks, such as autonomous driving [1, 2] and precision health [3, 4]. However, such heterogeneity between different views brings challenges for supervision signal (label) annotation, making the label scarcity problem widespread in practice. For example, diagnosing Alzheimer's Disease requires several physicians to jointly consider information from clinical records, Neuro-imaging scans, fluid biomarker readings, *etc.*, to make informed decisions [5, 6]. Leveraging extrinsic semantic information from data geometric structure, graph-based multi-view semi-supervised learning (GMvSSL) has been studied intensively and used in various applications [7–13]. In general, GMvSSL methods describe sample relationships using graphs and rely on the *smoothness assumption* [14]–that samples sharing the same label are likely to lie on the same manifold–to "propagate" label information across the graph.

Since existing GMvSSL approaches all focus on the extension of single-view semi-supervised learning methods to multi-view scenarios with late or early fusion strategies, most of them typically presume that all views are available for every sample. However, multi-view data in real applications frequently suffers from the miss of certain views due to machine failure or accessibility issues [15–18]. Learning

---

∗Corresponding author.

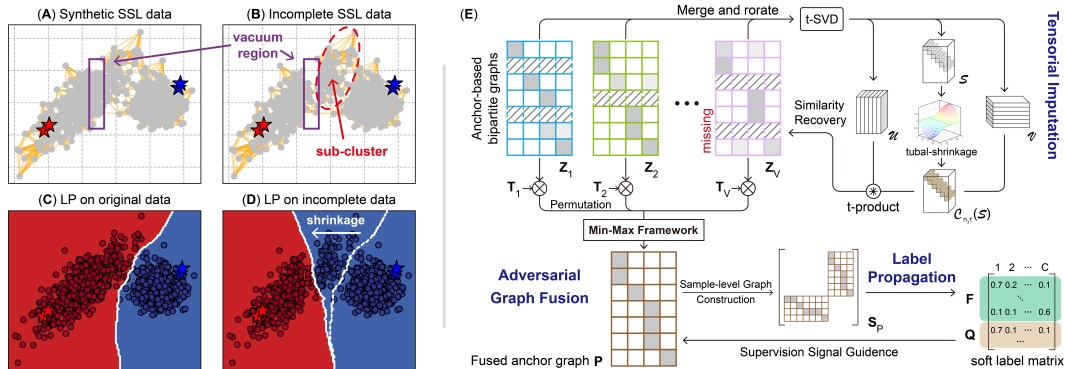

Figure 1: (A)–(D) is an example for the Sub-Cluster Problem (SCP), and (E) shows the proposed **AGF-TI**. SCP: sub-clusters caused by the missing samples break the smoothness assumption in Label Propagation (LP). To address SCP, **AGF-TI** comprises an adversarial graph fusion operator to learn a robust fused graph, and tensor learning to recover similarity relationships of missing samples.

for such dual missing issue (*i.e.*, missing views and scarce labels) is crucial but rarely studied. To address this, most recently, Zhuge *et al.* [19] proposed an incomplete multi-view semi-supervised learning (IMvSSL) method, termed AMSC. AMSC first learns multiple basic label matrices via label propagation based on partial similarity graphs, which are constructed among existing samples in individual views. Then, AMSC integrates them into a consensus label matrix for prediction. Despite adopting the $p$-th root strategy for adaptive view fusion, AMSC overlooks the distortions caused by the view missing issue to the graph or manifold structure.

In this paper, we argue that the view missing issue will incur unreliable neighbor relationships, thus breaking the key *smoothness assumption* in label propagation (LP). As shown in Fig. 1 (A) and (B), missing samples in each view may generate multiple "vacuum regions" that fragment a complete category cluster into several *sub-clusters*, thereby distorting the smooth local structure on the common manifold. We term this phenomenon the **S**ub-**C**luster **P**roblem (SCP). Comparing Fig. 1 (C) and (D), one could observe that SCP impedes the propagation of red label information to its corresponding sub-cluster, erroneously making the decision boundary recede into the vacuum region. As a result, SCP would not only cause inappropriate graph fusion by misrepresenting the local structure, but also directly degrade the quality of basic label matrices in individual views. Besides, to the best of our knowledge, no existing framework has been proposed to mitigate the impact of SCP on GMvSSL.

To address SCP, we propose **AGF-TI**, a novel **A**dversarial **G**raph **F**usion-based IMvSSL method with **T**ensorial **I**mputation, as shown in Fig. 1 (E). In essence, **AGF-TI** incorporates three key innovations: (1) an adversarial graph fusion operator that fully explores complementary structure information across views through a min-max framework to learn a robust consensus graph for label propagation; (2) a tensorial imputation function that stacks view-specific similarity graphs into a third-order tensor to recover the similarity relationships of missing samples with high-order consistent correlations across views and tensor nuclear norm regularization; and (3) an anchor-based acceleration strategy that significantly reduces the computational cost associated with min-max optimization and tensor learning. To solve the proposed max-min-max optimization problem, we design a novel and efficient algorithm with theoretical convergence that incorporates the reduced gradient descent and the alternative direction method of multiplier. To verify the effectiveness of **AGF-TI**, we conduct empirical experiments on multiple public datasets with different view missing and label annotation ratios. Extensive experimental results show that **AGF-TI** outperforms existing competitors, exhibiting a more robust ability to tackle the dual missing issue. Our contributions are summarized as follows:

- We identify a novel challenge in GMvSSL with missing views, termed as the Sub-Cluster Problem, where missing samples may disrupt the core smoothness assumption, leading to inaccurate graph fusion and degraded performance of view-specific label prediction.

- We are the first to incorporate a min-max framework into GMvSSL to learn a robust consensus graph against the distorted local structure, enhancing fusion performance. Unlike existing IMvSSL methods that disregard missing samples when constructing graphs, we propose exploiting high-order consistent information to reconstruct incomplete local structures.

- We develop an efficient alternative optimization algorithm with the reduced gradient descent to solve the intractable objective of **AGF-TI**. Empirical results on various public datasets validate the effectiveness and efficiency of our proposed **AGF-TI**.

## 2 Preliminary and Related Work

### 2.1 Graph-based Multi-view Semi-supervised Learning

In general, graph-based multi-view semi-supervised learning (GMvSSL) can be viewed as the combination of a graph-based semi-supervised learning (GSSL) and a multi-view fusion strategy. Assume that $\mathbf{X}=[\mathbf{x}_1,\ldots,\mathbf{x}_n]^\top\in\mathbb{R}^{n\times d}$ is a $d$ dimensional single-view data matrix with $n$ samples, where the first $\ell(\ll n)$ samples are labeled into $c$ classes as $\{y_i\in[c]\}_{i=1}^\ell$. For the sake of calculation, a one-hot label matrix $\mathbf{Y}\in\{0,1\}^{n\times c}$ is used to describe the situation of labels, *i.e.*, $Y_{ij}=1$ iff $y_i=j(i\leq\ell)$ and $Y_{ij}=0$ otherwise. Then, the GSSL can be solved by minimizing the cost function:

$$\mathcal{L}(\mathbf{F},\mathbf{L}_S) = \mathrm{Tr}(\mathbf{F}^\top\mathbf{L}_S\mathbf{F}) + \mathrm{Tr}\left((\mathbf{F}-\mathbf{Y})^\top\mathbf{B}(\mathbf{F}-\mathbf{Y})\right), \tag{1}$$

where $\mathbf{F}\in\mathbb{R}^{n\times c}$ is a soft label matrix, $\mathbf{B}$ is a diagonal regularization matrix, and $\mathbf{L}_S=\mathbf{D}_S-\mathbf{S}$ is a Laplacian matrix associated with $\mathbf{S}$. Here, matrix $\mathbf{S}\in[0,1]^{n\times n}$ is the symmetric similarity matrix where its element $\mathbf{S}_{ij}=\mathbf{S}_{ji}$ describes the similarity between $\mathbf{x}_i$ and $\mathbf{x}_j$. The matrix $\mathbf{D}_S$ is a diagonal degree matrix with entries $d_{ii}=\sum_t\mathbf{S}_{it}$. The cost function was explained with the *smoothness rule* and the *fitting rule* [14]. The left-hand term, named *smoothness rule*, plays a crucial role in mining extrinsic supervision signals from graph structure, forcing the soft label vectors of nearby samples to barely differ. However, the performance of the *smoothness rule* heavily depends on the smoothness assumption, which can be easily violated by SCP arising from incompleteness in multi-view scenarios.

Multi-view fusion strategy aims to exploit the complementary information from multiple views to improve the final performance. Among them, late fusion [7, 19] approaches integrate information at the decision level by leveraging multiple base label matrices obtained via GSSL in individual views, while early fusion [9, 10, 20] approaches operate graph fusion on view-specific graphs to learn a consensus one. Although most late fusion methods adopt a weighting strategy to balance view quality, they often overlook geometric consistencies across views, making it struggle to recognize graph distortions induced by missing samples, *i.e.*, SCP. Therefore, in this work, we focus on the early fusion approaches and propose a novel adversarial graph fusion method using a min-max framework to explore the complementary local structure to alleviate the impact of SCP.

### 2.2 Accelerated GMvSSL with Bipartite Graphs

Despite the promising performance of existing GMvSSL methods, the high computational complexity prevents their application to large-scale tasks. Assume that $\{\mathbf{X}_v\in\mathbb{R}^{n\times d_v}\}_{v=1}^V$ is a multi-view dataset with $n$ samples and $V$ views. The time and space complexity of most GMvSSL are $\mathcal{O}(n^3)$ and $\mathcal{O}(Vn^2)$ cubic and quadratic to sample numbers [9, 10]. To address this, GMvSSL accelerated by anchored or bipartite graphs has been widely studied [11, 19, 21]. These approaches effectively reduce the computational costs by constructing bipartite graphs with $m(\ll n)$ anchors selected by $k$-means [22, 23], BKHK [24], *etc.*, instead of entire samples. After anchor selection, the bipartite graphs in each view $\mathbf{Z}_v\in\mathbb{R}^{n\times m}$ can be effectively constructed by solving the following problem:

$$\min_{\mathbf{Z}_v}\sum_{i=1}^n\sum_{j=1}^m\left(\|\mathbf{x}_i^{(v)}-\mathbf{a}_j^{(v)}\|_2^2 Z_{ij}^{(v)} + \gamma(Z_{ij}^{(v)})^2\right), \text{ s.t. } \mathbf{Z}_v\mathbf{1}_m=\mathbf{1}_n, \mathbf{Z}_v\geq0 \tag{2}$$

where $\mathbf{x}_i^{(v)}$ and $\mathbf{a}_j^{(v)}$ represents the $i$-th sample and $j$-th anchor in the $v$-th view, respectively, $Z_{ij}^{(v)}$ is the $(i,j)$-th element of $\mathbf{Z}_v$, and $\mathbf{1}_n$ is an all-ones column vector with $n$ elements. Compared to the classical Gaussian kernel method [7], once the neighbor number $k$ is given, the model in Eq. (2) enjoys a better construction performance with a parameter-free closed-form solution [25]. After obtaining the bipartite graphs, we typically construct the sample-level graphs using $\mathbf{S}_v=\mathbf{Z}_v\mathbf{Z}_v^\top$ as the input of GMvSSL [26, 27]. Accelerated by Woodbury matrix identity, the time and space complexity could be reduced to $\mathcal{O}(nm^2)$ and $\mathcal{O}(Vnm)$, which can be applied to large-scale datasets effectively.

### 2.3 Third-order Tensor for Multi-view Learning

To exploit the high-order correlation among multiple views, tensor-based multi-view learning approaches have emerged [28, 29]. Specifically, these methods construct a third-order tensor by stacking

view-specific representation matrices and impose low-rank constraints to capture inter-view consistent structure. For example, Zhang *et al.* [30] apply a slice-based nuclear norm to model the high-order correlation. To better constrain the low-rankness of the multi-view tensor, recent works [31, 32] introduce a tensor average rank with the Tensor Nuclear Norm (TNN) based on the tensor Singular Value Decomposition (t-SVD), serving as the tightest convex approximation. Aiming to alleviate SCP, this work employs the third-order tensor with the TNN to leverage the high-order correlation for recovering the local structure of missing samples on the bipartite graphs. We give the definition of Tensor Nuclear Norm as follows. Basic tensor operators are defined in Appendix A.

**Definition 1 (Tensor Nuclear Norm)** [33] *Given a tensor $\mathcal{Z}\in\mathbb{R}^{n_1\times n_2\times n_3}$, the nuclear norm of the tensor is defined as $\|\mathcal{Z}\|_{\circledast}=\frac{1}{n_3}\sum_{k=1}^{n_3}\sum_{i=1}^{\min(n_1,n_2)}\mathcal{S}_f^k(i,i)$, where $\mathcal{S}_f$ is obtained by t-SVD of $\mathcal{Z}=\mathcal{U}*\mathcal{S}*\mathcal{V}^\top$ in Fourier domain through the fast Fourier transform $\mathcal{S}_f=fft(\mathcal{S},[],3)$, and $\mathcal{S}_f^k$ is the k-th frontal slice of $\mathcal{S}_f$.*

# 3 Methodology

## 3.1 Proposed Formulation

Assume that $\pi_v$ and $\omega_v$ collect the index of existing and missing samples in the $v$-th view, respectively. Thus, the bipartite graph $\mathbf{Z}_v$ can be divided into $\mathbf{Z}_{\pi_v}$ and $\mathbf{Z}_{\omega_v}$, where $\mathbf{Z}_{\pi_v}$ could be constructed by Eq. (2) among existing samples while $\mathbf{Z}_{\omega_v}$ is initialized as equal probability matrix. Different to previous works [10, 12] that fuse the sample-level graphs $\mathbf{S}_v\in\mathbb{R}^{n\times n}$, we aim to directly obtain a consensus anchored graph $\mathbf{P}\in\mathbb{R}^{n\times m}$ based on $\{\mathbf{Z}_v\}_{v=1}^V$, which accelerates subsequent label propagation and tensor learning. Rethinking Eq. (2), it independently constructs the bipartite graph in each view, and unavoidably introduces the anchor-unaligned problem [34], degrading fusion performance. To tackle this issue, we introduce the permutation matrices $\{\mathbf{T}_v\in\mathbb{R}^{m\times m}\}_{v=1}^V$ to align the anchors between different views. Besides, to adaptively emphasize the contributions made by various views to the fused graph, we assign a learnable weight $\alpha_v$ ($v\in[V]$) to each view. Finally, inspired by adversarial training, we design a novel Adversarial Graph Fusion (AGF) operator based on a min-max framework, which is defined as follows:

$$\min_{\boldsymbol{\alpha}\in\Delta_V^1}\max_{\mathbf{P}\in\Delta_n^m}\texttt{AGF}(\mathbf{P},\{\mathbf{Z}_v\}_{v=1}^V)\triangleq\sum_{v=1}^V\alpha_v^2\mathrm{Tr}\left(\mathbf{P}^\top(\mathbf{Z}_v\mathbf{T}_v)\right)-\beta\|\mathbf{P}\|_F^2, \text{ s.t. } \mathbf{T}_v^\top\mathbf{T}_v=\mathbf{I}_m\ (\forall v), \quad (3)$$

where $\mathbf{I}_m\in\mathbb{R}^{m\times m}$ is an identity matrix, $\boldsymbol{\alpha}=[\alpha_1,\ldots,\alpha_V]^\top$ and $\Delta_n^m=\{\zeta\in\mathbb{R}^{n\times m}|\zeta\mathbf{1}_m=\mathbf{1}_n,\zeta\geq 0\}$.

**Remark 1 (Benefits of AGF)** *Compared to prior early fusion methods, AGF has the following merits: (1) AGF directly learns an bipartite graph, which improves the efficiency of the algorithm; (2) Regularization by the min-max framework of $\boldsymbol{\alpha}$ and $\mathbf{P}$ makes the model less sensitive to minor data fluctuations, e.g., sub-clusters, alleviating SCP and generating a more robust consensus graph [35].*

Furthermore, to capture high-order correlations across views, we stack the bipartite graphs $\{\mathbf{Z}_v\}_{v=1}^V$ into a tensor $\mathcal{Z}=\Phi(\mathbf{Z}_1,\ldots,\mathbf{Z}_V)\in\mathbb{R}^{m\times V\times n}$, where $\Phi(\cdot)$ is a merging and rotating operator [36]. By optimizing $\mathcal{Z}$ with TNN, the similarity relationships between missing samples and anchors in each view are imputed via the cross-view consistency information. In this way, the incomplete graph structures could be recovered, thus further alleviating the effect of SCP.

Finally, to enhance the semantic information of unlabeled data, we construct a sample-level graph for label propagation by using limited labeled instances. To better leverage the local structural information across neighbors, following [11], we construct the graph among fused sample and anchor nodes as $\mathbf{S}_P=\begin{bmatrix} & \mathbf{P} \\ \mathbf{P}^\top & \end{bmatrix}\in\mathbb{R}^{(n+m)\times(n+m)}$. By introducing a soft label matrix of fused anchors as $\mathbf{Q}\in\mathbb{R}^{m\times c}$, the loss function in Eq. (1) can be equally rewritten into a performance gain form with the normalized Laplacian matrix $\tilde{\mathbf{L}}_{S_P}=\mathbf{I}_{n+m}-\mathbf{D}_{S_P}^{-\frac{1}{2}}\mathbf{S}_P\mathbf{D}_{S_P}^{-\frac{1}{2}}=\mathbf{I}_{n+m}-\hat{\mathbf{S}}_P$:

$$\mathcal{R}(\hat{\mathbf{F}},\tilde{\mathbf{L}}_{S_P})=\mathrm{Tr}\left(\hat{\mathbf{F}}^\top\hat{\mathbf{S}}_P\hat{\mathbf{F}}\right)+2\mathrm{Tr}\left(\hat{\mathbf{B}}\hat{\mathbf{Y}}\hat{\mathbf{F}}^\top\right)-\mathrm{Tr}\left((\mathbf{I}_{n+m}+\hat{\mathbf{B}})\hat{\mathbf{F}}^\top\hat{\mathbf{F}}\right), \quad\quad (4)$$

where $\hat{\mathbf{F}}=[\mathbf{F};\mathbf{Q}]\in\mathbb{R}^{(n+m)\times c}$, $\hat{\mathbf{Y}}=[\mathbf{Y};\mathbf{0}]$, and $\hat{\mathbf{B}}$ is a diagonal matrix with the $i$-th entry being regularization parameter. Therefore, the final objective of **AGF-TI** can be expressed as:

$$
\begin{cases}
\displaystyle\max_{\{\mathbf{Z}_{\omega_v},\mathbf{T}_v\}_{v=1}^V,\hat{\mathbf{F}}}\min_{\boldsymbol{\alpha}}\max_{\mathbf{P}}\ \mathcal{R}(\hat{\mathbf{F}},\tilde{\mathbf{L}}_{S_P})+\lambda\mathrm{AGF}(\mathbf{P},\{\mathbf{Z}_v\}_{v=1}^V)-\rho\|\mathcal{Z}\|_{\circledast}\\
\text{s.t. }\mathbf{P},\mathbf{Z}_v\in\Delta_n^m,\mathbf{T}_v^\top\mathbf{T}_v=\mathbf{I}_m\ (\forall v\in[V]),\boldsymbol{\alpha}\in\Delta_V^1,\mathcal{Z}=\Phi(\mathbf{Z}_1,\ldots,\mathbf{Z}_V),
\end{cases}
\tag{5}
$$

where $\lambda$ and $\rho$ are nonnegative regularization parameters.

### 3.2 Optimization

To solve the intractable max-min-max model in Eq. (5), we propose an efficient solution by combining the reduced gradient descent and alternative direction method of multipliers (ADMM). Inspired by ADMM, we introduce an auxiliary tensor variable $\mathcal{G}$ to relax $\mathcal{Z}$, so the objective function of Eq. (5) can be rewritten as the following augmented Lagrangian function:

$$
\begin{aligned}
\mathcal{J}(\{\mathbf{Z}_{\omega_v},\mathbf{T}_v\}_v^V,\hat{\mathbf{F}},\boldsymbol{\alpha},\mathbf{P},\mathcal{G},\mathcal{W})=&\mathrm{Tr}\left(\hat{\mathbf{F}}^\top\hat{\mathbf{S}}_P\hat{\mathbf{F}}\right)+2\mathrm{Tr}\left(\hat{\mathbf{B}}\hat{\mathbf{Y}}\hat{\mathbf{F}}^\top\right)-\mathrm{Tr}\left((\mathbf{I}_{n+m}+\hat{\mathbf{B}})\hat{\mathbf{F}}^\top\hat{\mathbf{F}}\right)\\
&+\lambda\sum_v\alpha_v^2\mathrm{Tr}\left(\mathbf{P}^\top(\mathbf{Z}_v\mathbf{T}_v)\right)-\beta_\lambda\|\mathbf{P}\|_F^2-\rho\|\mathcal{G}\|_{\circledast}-\langle\mathcal{W},\mathcal{Z}-\mathcal{G}\rangle-\frac{\eta}{2}\|\mathcal{Z}-\mathcal{G}\|_F^2,
\end{aligned}
\tag{6}
$$

where $\beta_\lambda=\lambda\cdot\beta$, $\mathcal{W}$ is Lagrange Multiplier, and $\eta>0$ serves as a penalty parameter to control convergence. Then, the optimization problem of Eq. (6) can be decomposed into five subproblems, each of which optimizes its respective variables independently while keeping others fixed.

● $\mathbf{Z}_{\omega_v}$**-Subproblem:** Fixing the other variables, the problem in Eq. (6) can be disassembled into $V$ separate subproblems *w.r.t.* $\mathbf{Z}_{\omega_v}$, $v=1,2,\ldots V$:

$$
\min_{\mathbf{Z}_{\omega_v}}\frac{\eta}{2}\|\mathbf{Z}_v-\mathbf{G}_v\|_F^2+\langle\mathbf{W}_v,\mathbf{Z}_v-\mathbf{G}_v\rangle-\lambda\alpha_v^2\mathrm{Tr}\left(\mathbf{P}^\top(\mathbf{Z}_v\mathbf{T}_v)\right),\ \text{s.t. }\mathbf{Z}_{\omega_v}\in\Delta_{|\omega_v|}^m.
\tag{7}
$$

The problem in Eq. (7) can be further rewritten as the following element-wise form:

$$
\min_{\mathbf{Z}_{\omega_v}\in\Delta_{|\omega_v|}^m}\sum_{i\in\omega_v}\sum_{j=1}^m\frac{\eta}{2}(Z_{ij}^{(v)}-G_{ij}^{(v)})^2+W_{ij}^{(v)}(Z_{ij}^{(v)}-G_{ij}^{(v)})-\lambda\alpha_v^2(\mathbf{P}\mathbf{T}_v^\top)_{ij}Z_{ij}^{(v)}.
\tag{8}
$$

Noting that the problem in Eq. (8) is independent between different $i$, so we can individually solve the following problem in vector form for each row of $\mathbf{Z}_{\omega_v}$:

$$
\min_{\mathbf{Z}_{i\cdot}^{(v)}}\left\|\mathbf{Z}_{i\cdot}^{(v)}-\left(\mathbf{G}_{i\cdot}^{(v)}-\frac{1}{\eta}\left(\mathbf{W}_{i\cdot}^{(v)}-\lambda\alpha_v^2(\mathbf{P}\mathbf{T}_v)_{i\cdot}\right)\right)\right\|_2^2,\ \text{s.t. }\mathbf{Z}_{i\cdot}^{(v)}\mathbf{1}_m=1,\mathbf{Z}_{i\cdot}^{(v)}\geq0,\ i\in\omega_v,
\tag{9}
$$

which can be solved with a closed-form solution [25].

● $\mathbf{P}$ **and** $\boldsymbol{\alpha}$**-Subproblem:** By fixing the other variables, we derive a min-max optimization problem *w.r.t.* $\mathbf{P}$ and $\boldsymbol{\alpha}$, and then rewrite it with an optimal value function of the inner maximization problem:

$$
\min_{\boldsymbol{\alpha}\in\Delta_V^1}h(\boldsymbol{\alpha}),\quad h(\boldsymbol{\alpha})\triangleq\max_{\mathbf{P}\in\Delta_n^m}\lambda\sum_v\alpha_v^2\mathrm{Tr}(\mathbf{P}^\top(\mathbf{Z}_v\mathbf{T}_v))-\beta_\lambda\|\mathbf{P}\|_F^2-\mathrm{Tr}(\hat{\mathbf{F}}^\top\tilde{\mathbf{L}}_{S_P}\hat{\mathbf{F}}).
\tag{10}
$$

Recall that $\hat{\mathbf{F}}=[\mathbf{F};\mathbf{Q}]$, based on the property of the normalized Laplacian matrix, the relationship $\mathrm{Tr}(\hat{\mathbf{F}}^\top\tilde{\mathbf{L}}_{S_P}\hat{\mathbf{F}})=\sum_i^n\sum_j^m\|\mathbf{F}_{i\cdot}./\sqrt{d_i}-\mathbf{Q}_{j\cdot}./\sqrt{d_{n+j}}\|_2^2P_{ij}$ holds, where $d_i=\sum_{k=1}^{n+m}\mathbf{S}_P(i,k)$. Similar to the subproblem of $\mathbf{Z}_{\omega_v}$, we can obtain the optimal $\mathbf{P}^\star$ of the inner maximization problem by solving the following proximal problem for each row of $\mathbf{P}$ with a closed-form solution:

$$
\mathbf{P}_{i\cdot}^\star=\arg\min_{\mathbf{P}_{i\cdot}}\left\|\mathbf{P}_{i\cdot}-\frac{1}{2\beta_\lambda}(\lambda\tilde{\mathbf{Z}}_{i\cdot}-\mathbf{H}_{i\cdot})\right\|_2^2,\ \text{s.t. }\mathbf{P}_{i\cdot}\mathbf{1}_m=1,\mathbf{P}_{i\cdot}\geq0,
\tag{11}
$$

where $\tilde{\mathbf{Z}}=\sum_v\alpha_v^2\mathbf{Z}_v\mathbf{T}_v$ is the weighted bipartite graph and $\mathbf{H}\in\mathbb{R}^{n\times m}$ measures the distance between the soft labels of $\mathbf{F}$ and $\mathbf{Q}$ with its element $H_{ij}=\|\mathbf{F}_{i\cdot}./\sqrt{d_i}-\mathbf{Q}_{j\cdot}./\sqrt{d_{n+j}}\|_2^2$. Since the feasible region of Eq. (11) is a closed convex set and its objective function is strictly convex, the Hilbert projection theorem guarantees the uniqueness of the optimal solution $\mathbf{P}^\star$. Then, leveraging Theorem 4.1 in [37], we have the following theorem with its proof provided in Appendix C.

**Theorem 1** $h(\boldsymbol{\alpha})$ *is differentiable, and its gradient can be calculated as* $\frac{\partial h(\boldsymbol{\alpha})}{\partial \alpha_v} = 2\lambda\alpha_v Tr(\mathbf{P}^{\star\top}\mathbf{Z}_v\mathbf{T}_v)$, *where* $\mathbf{P}^\star$ *is the optimal solution of the inner maximization problem.*

Therefore, a reduced gradient descent method can be developed to solve the optimization problem in Eq. (10). Specifically, we first calculate the gradient of $h(\boldsymbol{\alpha})$ by Theorem 1 and update the $\boldsymbol{\alpha}$ along the direction of the gradient descent over the simplex constraint $\boldsymbol{\alpha} \in \Delta_V^1$ with the optimal $\mathbf{P}^\star$.

Consider the equality constraint of $\boldsymbol{\alpha}$, supposing that $\alpha_v$ is a non-zero entry of $\boldsymbol{\alpha}$ and $\nabla h(\boldsymbol{\alpha})$ is the reduced gradient of $h(\boldsymbol{\alpha})$. Following [38, 39], the $v$-th entry of the reduced gradient can be constructed as follows:

$$[\nabla h(\boldsymbol{\alpha})]_v = \frac{\partial h(\boldsymbol{\alpha})}{\partial \alpha_v} - \frac{\partial h(\boldsymbol{\alpha})}{\partial \alpha_u}, \ \forall v \neq u, \quad [\nabla h(\boldsymbol{\alpha})]_u = \sum_{v=1,v\neq u}^{V} \left( \frac{\partial h(\boldsymbol{\alpha})}{\partial \alpha_u} - \frac{\partial h(\boldsymbol{\alpha})}{\partial \alpha_v} \right), \quad (12)$$

where $u$ is typically set as the index of the largest entry of $\boldsymbol{\alpha}$, leading to better numerical stability [40]. Since $-\nabla h(\boldsymbol{\alpha})$ represents a descent direction to minimize $h(\boldsymbol{\alpha})$, we can directly set the descent direction $\mathbf{g} = -\nabla h(\boldsymbol{\alpha})$. To ensure the non-negativity of $\boldsymbol{\alpha}$, we further modify the descent direction $g_v$ to zero iff $\alpha_v = 0$ and $[\nabla h(\boldsymbol{\alpha})]_v > 0$. Then, $\boldsymbol{\alpha}$ can be updated using the rule of $\boldsymbol{\alpha} \leftarrow \boldsymbol{\alpha} + \theta\mathbf{g}$, where $\theta$ is the optimal step length calculated by a linear search mechanism, *e.g.*, Armijo's rule. The detailed procedure for solving the problem in Eq. (10) is outlined in Algorithm 1 in Appendix B.2.

• $\hat{\mathbf{F}}$-**Subproblem:** Fixing the other variables leads to the following problem for $\hat{\mathbf{F}}$,

$$\max_{\hat{\mathbf{F}}} \ \mathcal{R}(\hat{\mathbf{F}}, \tilde{\mathbf{L}}_{S_P}), \ \text{s.t.} \ \hat{\mathbf{F}} = [\mathbf{F}; \mathbf{Q}] \in \mathcal{R}^{(n+m)\times c}. \tag{13}$$

By setting the derivative of $\mathcal{R}(\hat{\mathbf{F}}, \tilde{\mathbf{L}}_{S_P})$ to zero, $\hat{\mathbf{F}}$ can be updated as $\hat{\mathbf{F}}^\star = (\tilde{\mathbf{L}}_{S_P} + \hat{\mathbf{B}})^{-1}\hat{\mathbf{B}}\hat{\mathbf{Y}}$. Since $\mathbf{L}_{\hat{S}_P}$ and $\hat{\mathbf{B}}$ are both block matrices, we can adopt the blockwise inversion to solve the first term:

$$(\tilde{\mathbf{L}}_{S_P} + \hat{\mathbf{B}})^{-1} = \begin{bmatrix} \mathbf{I}_n + \mathbf{B}_n & -\mathbf{P}\Lambda^{-\frac{1}{2}} \\ -\Lambda^{-\frac{1}{2}}\mathbf{P}^\top & \mathbf{I}_m + \mathbf{B}_m \end{bmatrix}^{-1} = \begin{bmatrix} \mathbf{C}_1^{-1} & -\mathbf{M}_{11}^{-1}\mathbf{M}_{12}\mathbf{C}_2^{-1} \\ -\mathbf{C}_2^{-1}\mathbf{M}_{21}\mathbf{M}_{11}^{-1} & \mathbf{C}_2^{-1} \end{bmatrix}, \quad (14)$$

where $\Lambda \in \mathbb{R}^{m\times m}$ is a diagonal matrix with element $\Lambda_{jj} = \sum_i^n P_{ij}$, $\mathbf{M}_{11} = \mathbf{I}_n + \mathbf{B}_n$, $\mathbf{M}_{12} = -\mathbf{P}\Lambda^{-\frac{1}{2}}$, $\mathbf{M}_{21} = -\Lambda^{-\frac{1}{2}}\mathbf{P}^\top$, $\mathbf{M}_{22} = \mathbf{I}_m + \mathbf{B}_m$, $\mathbf{C}_1 = \mathbf{M}_{11} - \mathbf{M}_{12}\mathbf{M}_{22}^{-1}\mathbf{M}_{21}$, and $\mathbf{C}_2 = \mathbf{M}_{22} - \mathbf{M}_{21}\mathbf{M}_{11}^{-1}\mathbf{M}_{12}$. Since $\mathbf{C}_1 \in \mathbb{R}^{n\times n}$ needs time complexity $\mathcal{O}(n^3)$ to solve $\mathbf{C}_1^{-1}$, we utilize the Woodbury matrix identity to accelerate the inversion by $\mathbf{C}_1^{-1} = \mathbf{M}_{11}^{-1} + \mathbf{M}_{11}^{-1}\mathbf{M}_{12}(\mathbf{M}_{22} - \mathbf{M}_{21}\mathbf{M}_{11}^{-1}\mathbf{M}_{12})^{-1}\mathbf{M}_{21}\mathbf{M}_{11}^{-1}$, which reduces $\mathcal{O}(n^3)$ to $\mathcal{O}(nm^2)$. Note that $\mathbf{M}_{11} \in \mathbb{R}^{n\times n}$, but it is a diagonal matrix and its inversion can be easily obtained with $\mathcal{O}(n)$. Then, leveraging the blockwise inversion in Eq. (14), the update formula for $\hat{\mathbf{F}}^\star = [\mathbf{F}^\star; \mathbf{Q}^\star]$ can be divided into two parts *w.r.t.* $\mathbf{F}$ and $\mathbf{Q}$, respectively:

$$\begin{cases} \mathbf{F}^\star = \mathbf{M}_{11}^{-1}\mathbf{M}_{12}(\mathbf{M}_{22} - \mathbf{M}_{21}\mathbf{M}_{11}^{-1}\mathbf{M}_{12})^{-1}\mathbf{M}_{21}\mathbf{M}_{11}^{-1}\mathbf{B}_n\mathbf{Y} + \mathbf{M}_{11}^{-1}\mathbf{B}_n\mathbf{Y}, \\ \mathbf{Q}^\star = -(\mathbf{M}_{22} - \mathbf{M}_{21}\mathbf{M}_{11}^{-1}\mathbf{M}_{12})^{-1}\mathbf{M}_{21}\mathbf{M}_{11}^{-1}\mathbf{B}_n\mathbf{Y}. \end{cases} \tag{15}$$

• $\mathcal{G}$-**Subproblem:** When other variables are fixed, the subproblem for $\mathcal{G}$ is formulated as:

$$\min_{\mathcal{G}} \ \frac{\rho}{\eta}\|\mathcal{G}\|_{\circledast} + \frac{1}{2}\left\|\mathcal{G} - \left(\mathcal{Z} + \frac{\mathcal{W}}{\eta}\right)\right\|_F^2. \tag{16}$$

The subproblem in Eq. (16) can be solved by the following theorem.

**Theorem 2** [36] *Suppose* $\mathcal{G}, \mathcal{F} \in \mathbb{R}^{n_1\times n_2\times n_3}$ *and* $\tau > 0$, *the globally optimal solution to* $\min_{\mathcal{G}} \tau\|\mathcal{G}\|_{\circledast} + \frac{1}{2}\|\mathcal{G} - \mathcal{F}\|_F^2$ *is given by the tensor tubal-shrinkage operator, i.e.,* $\mathcal{G} = \mathcal{C}_{n_3\tau}(\mathcal{F}) = \mathcal{U} * \mathcal{C}_{n_3\tau}(\mathcal{S}) * \mathcal{V}^\top$, *where* $\mathcal{F} = \mathcal{U} * \mathcal{S} * \mathcal{V}^\top$ *is obtained by t-SVD and* $\mathcal{C}_{n_3\tau}(\mathcal{S}) = \mathcal{S} * \mathcal{J}$. *Herein,* $\mathcal{J}$ *is an* $n_1 \times n_2 \times n_3$ *f-diagonal tensor whose diagonal element in the Fourier domain is* $\mathcal{J}_f^k(i,i) = (1 - n_3\tau/\mathcal{S}_f^k(i,i))_+$.

• $\mathbf{T}_v$-**Subproblem:** By fixing the other variables, $\mathbf{T}_v$ can be independently updated by,

$$\mathbf{T}_v^\star = \arg \max_{\mathbf{T}_v} \ Tr(\mathbf{T}_v^\top \mathbf{Z}_v^\top \mathbf{P}), \ \text{s.t.} \ \mathbf{T}_v^\top \mathbf{T}_v = \mathbf{I}_m. \tag{17}$$

The optimal solution $\mathbf{T}_v^\star$ of the problem (17) is $\mathbf{U}_v\mathbf{V}_v^\top$, where $\mathbf{U}_v$ and $\mathbf{V}_v$ are the left and right singular matrix of $\mathbf{Z}_v^\top\mathbf{P}$. Its detailed proof is provided in Appendix B.4.

At last, the Lagrange multiplier and penalty parameter are updated as $\mathcal{W}=\mathcal{W}+\eta(\mathcal{Z}-\mathcal{G})$ and $\eta=\min(\gamma_\eta\eta,\eta_{\max})$, respectively, where $\gamma_\eta>1$ is used to accelerate convergence.

The procedure of our method is summarized in Algorithm 2 in Appendix B.5. Besides, Appendix D.1 provides a theoretical convergence analysis, showing that the variable sequence obtained by Algorithm 2 converges to a stationary point. Its time and space complexity is discussed in Appendix D.2.

## 4  Experiments

### 4.1  Experimental Settings

**Datasets.** We conduct all experiments on six public datasets, including CUB, UCI-Digit, Caltech101-20, OutScene, MNIST-USPS, and AwA. The brief information of these datasets are presented in Table 1. More details of them are shown in Appendix E.1.

**Baselines.** To validate the effectiveness of **AGF-TI**, we compare it with the following methods. First, **SLIM** [46], and **AMSC** [19] serve as two state-of-the-art IMvSSL baselines. Furthermore, we include numerous GMvSSL algorithms, such as **AMMSS** [7], **AMGL** [8], **MLAN** [9], **AMUSE** [10], **FMSSL** [11], **FM-SEL** [12], and **CFMSC** [13]. We also incorporate popular regression-based MvSSL methods, *i.e.*, **MVAR** [47] and **ERL-MVSC** [48]. Since

Table 1: The description of six datasets

| Datasets | Samples | Views | Classes | Anchors |
|---|---|---|---|---|
| CUB [41] | 600 | 2 | 10 | 64 |
| UCI-Digit [42] | 2,000 | 3 | 10 | 256 |
| Caltech101-20 [41] | 2,386 | 6 | 20 | 256 |
| OutScene [43] | 2,688 | 4 | 8 | 256 |
| MNIST-USPS [44] | 5,000 | 2 | 10 | 256 |
| AwA [45] | 10,158 | 2 | 50 | 512 |

most baselines, except **SLIM** and **AMSC**, cannot handle the incomplete data, following [19], we adopt deep matrix factorization (DMF) [49] to recover feature matrices for a fair comparison. More details are shown in Appendix E.2.

**Parameter setting.** Following [11], we apply the BKHK algorithm to select $m$ anchor points among existing samples in each view. The number of selected anchors $m$ is presented in Table 1. Then we solve the problem in Eq. (2) to construct the bipartite graphs, *i.e.*, $\{\mathbf{Z}_{\pi_v}\}_{v=1}^V$ with the neighbor number $k$ set to 7. **AGF-TI** has three regularization parameters $\lambda$, $\beta_\lambda$, and $\rho$. In our experiments, $\lambda$ is fixed to $V^2$, while $\rho$ and $\beta_\lambda$ are tuned in $\{10^1,10^2,10^3\}$ and $\{2^0,\ldots,2^6\}$, respectively. We apply accuracy (ACC), precision (PREC), and F1-score (F1) as the evaluation metrics. Each experiment is independently conducted ten times, and the final average results are reported.

### 4.2  Main Results

To comprehensively evaluate the effectiveness of **AGF-TI**, we compare it with the baselines from two perspectives, *i.e.*, view missing and label scarcity. For the view missing issue, we follow [19] to randomly select VMR% (view missing ratio) examples in dataset as incomplete examples, which randomly miss 1~$V-1$ views. On this basis, we randomly select LAR% (label annotation ratio) examples of each class as labeled data to simulate the label scarcity setting.

From view missing perspective, Table 2 presents the comparison in metrics of different methods on six datasets under multiple VMRs (30%, 50%, 70%) when LAR% is 5%. From Table 2, we can observe that: (1) Compared to AMSC, tailored for IMvSSL, GMvSSL methods with DMF achieve comparable or even higher performance on some datasets like CUB and MNIST-USPS. This further validates our observation that missing samples disregarded by AMSC could distort local structure and mislead label propagation, thereby degrading performance. (2) **AGF-TI** consistently achieves the highest performance under multiple VMRs and metrics on all datasets except AwA, where it obtains two sub-optimal results at VMR=30% while showing dominant superiority with increased VMR. The results demonstrate that **AGF-TI** can effectively tackle the dual missing issue. (3) Compared with IMvSSL baselines, *i.e.*, SLIM and AMSC, which disregard missing samples, the tensorial imputation strategy makes **AGF-TI** more robust to incomplete multi-view data. For instance, on OutScene, as VMR is increased from 30% to 70%, the ACC decline of **AGF-TI** is less than 6%, while the decreases of SLIM and AMSC are nearly 13%. Detailed results with standard deviations are in Appendix F.1.

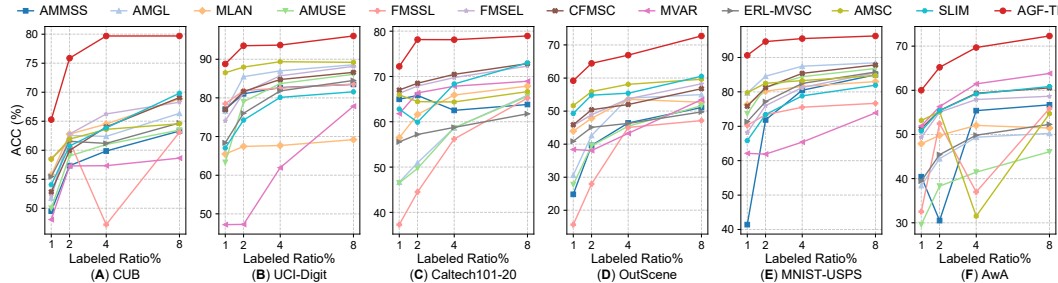

Figure 2: ACC results on six datasets with LAR varying in {1%, 2%, 4%, 8%} when VMR is 60%.

From label scarcity perspective, Fig. 2 presents the ACC results under different LARs varied in {1%, 2%, 4%, 8%} with VMR fixed to 60%. Compared with early fusion-based GMvSSL approaches, *i.e.*, MLAN, AMUSE, FMSEL and CFMSC, **AGF-TI** consistently outperforms them over all cases. This result suggests that the consensus graph fused through adversarial graph fusion in **AGF-TI** can effectively alleviate the negative impact of structural distortions caused by missing samples on label propagation. Besides, we observe that the performance of **AGF-TI** when LAR is only 1% is on par with most baselines under higher LAR, indicating higher label utilization efficiency of our method.

### 4.3 Ablation Study

**AGF-TI** contains two main parts: adversarial graph fusion and tensorial imputation. To validate its effectiveness, we conduct the following ablation studies under different VMRs when LAR is fixed to 5%. In the adversarial graph fusion part, we remove the permutation matrix for anchor alignment and the weight coefficient for view quality assessment, marked as "w/o $\mathbf{T}_v$" and "w/o $\alpha_v$", respectively. To evaluate the tensorial imputation part, we remove the update of $\mathbf{Z}_{\omega_v}$ from the optimization algorithm, termed as "w/o TI". The ablation results are listed in Table 3. From the results,

Table 2: Mean results of compared methods on six datasets under different VMRs (view missing ratios) when fix LAR (label annotation ratio) to 5%.

| Method | CUB | | | | | | | | | UCI-Digit | | | | | | | | |
|---|---|---|---|---|---|---|---|---|---|---|---|---|---|---|---|---|---|---|
| | VMR=30% | | | VMR=50% | | | VMR=70% | | | VMR=30% | | | VMR=50% | | | VMR=70% | | |
| | ACC | PREC | F1 | ACC | PREC | F1 | ACC | PREC | F1 | ACC | PREC | F1 | ACC | PREC | F1 | ACC | PREC | F1 |
| AMMSS | 67.33 | 67.33 | 64.57 | 48.44 | 48.44 | 44.25 | 62.74 | 62.74 | 60.92 | 91.46 | 91.46 | 91.43 | 87.30 | 87.30 | 87.24 | 76.04 | 76.04 | 75.87 |
| AMGL | 67.11 | 67.11 | 64.64 | 65.77 | 65.77 | 63.32 | 61.75 | 61.75 | 60.04 | 91.06 | 91.06 | 90.93 | 89.73 | 89.73 | 89.65 | 85.32 | 85.32 | 85.22 |
| MLAN | 70.09 | 70.09 | 68.35 | 67.75 | 67.75 | 66.04 | 62.25 | 62.25 | 60.68 | 86.13 | 86.13 | 86.27 | 74.82 | 74.82 | 76.31 | 62.43 | 62.43 | 65.09 |
| AMUSE | 64.26 | 64.26 | 61.84 | 63.21 | 63.21 | 61.35 | 62.77 | 62.77 | 60.90 | 90.61 | 90.61 | 90.50 | 87.21 | 87.21 | 87.13 | 81.91 | 81.91 | 81.81 |
| FMSSL | 65.30 | 65.30 | 62.31 | 48.72 | 45.72 | 44.62 | 64.49 | 64.49 | 62.68 | 91.12 | 91.12 | 91.07 | 86.72 | 86.72 | 86.62 | 78.00 | 78.00 | 77.72 |
| FMSEL | 70.54 | 70.54 | 69.21 | 67.28 | 67.28 | 66.16 | 62.35 | 62.35 | 60.81 | 92.19 | 92.19 | 92.16 | 88.99 | 88.99 | 88.95 | 83.53 | 83.53 | 83.40 |
| CFMSC | 71.65 | 71.65 | 70.81 | 69.19 | 69.19 | 68.02 | 55.23 | 54.23 | 53.13 | 91.48 | 91.48 | 91.44 | 88.14 | 88.14 | 88.08 | 83.47 | 83.47 | 83.33 |
| MVAR | 66.49 | 66.49 | 65.19 | 61.51 | 61.51 | 60.20 | 50.18 | 50.18 | 41.15 | 80.51 | 80.51 | 80.43 | 73.46 | 73.46 | 73.11 | 68.01 | 68.01 | 67.78 |
| ERL-MVSC | 65.75 | 65.75 | 65.48 | 61.51 | 61.51 | 61.34 | 59.98 | 59.98 | 59.73 | 86.80 | 86.80 | 86.78 | 84.61 | 84.61 | 84.59 | 79.20 | 79.20 | 79.20 |
| AMSC | 68.75 | 68.75 | 66.14 | 67.60 | 67.60 | 65.95 | 64.39 | 64.39 | 63.08 | 93.87 | 93.87 | 93.86 | 91.21 | 91.21 | 91.19 | 87.59 | 87.59 | 87.55 |
| SLIM | 68.30 | 68.30 | 67.36 | 65.12 | 65.12 | 63.48 | 64.04 | 64.04 | 63.42 | 84.93 | 84.93 | 84.80 | 81.41 | 81.41 | 81.41 | 72.55 | 72.55 | 72.43 |
| **Ours** | **78.33** | **78.33** | **77.03** | **80.23** | **80.23** | **79.10** | **74.25** | **74.25** | **72.30** | **95.98** | **95.98** | **95.97** | **95.23** | **95.23** | **95.25** | **95.16** | **95.16** | **95.13** |

| Method | Caltech101-20 | | | | | | | | | OutScene | | | | | | | | |
|---|---|---|---|---|---|---|---|---|---|---|---|---|---|---|---|---|---|---|
| AMMSS | 69.20 | 30.41 | 32.19 | 65.87 | 25.59 | 26.30 | 60.87 | 20.39 | 20.57 | 58.11 | 55.68 | 53.77 | 53.47 | 51.16 | 49.54 | 44.43 | 42.16 | 40.56 |
| AMGL | 61.85 | 22.88 | 24.97 | 62.18 | 22.50 | 24.06 | 61.60 | 21.51 | 22.56 | 58.28 | 57.71 | 58.32 | 56.19 | 55.58 | 55.54 | 48.33 | 47.84 | 46.69 |
| MLAN | 74.07 | 39.74 | 43.49 | 69.92 | 33.26 | 36.07 | 63.60 | 25.42 | 26.95 | 64.08 | 63.89 | 63.85 | 57.11 | 57.00 | 56.81 | 43.57 | 42.90 | 42.04 |
| AMUSE | 60.52 | 21.87 | 24.22 | 61.85 | 22.27 | 23.95 | 60.42 | 20.55 | 21.40 | 56.80 | 56.11 | 57.04 | 55.17 | 54.64 | 54.80 | 42.76 | 40.95 | 38.90 |
| FMSSL | 69.28 | 30.60 | 30.36 | 43.20 | 10.51 | 9.87 | 57.01 | 15.71 | 15.11 | 52.88 | 51.34 | 50.16 | 48.73 | 47.52 | 46.61 | 42.76 | 40.95 | 38.90 |
| FMSEL | 77.86 | 47.16 | 51.59 | 72.63 | 37.77 | 41.09 | 66.92 | 29.27 | 31.49 | 66.10 | 66.03 | 66.10 | 56.42 | 56.17 | 56.46 | 50.34 | 49.54 | 49.14 |
| CFMSC | 78.89 | 49.39 | 53.70 | 73.46 | 39.55 | 43.15 | 67.76 | 31.11 | 33.90 | 67.41 | 67.07 | 66.84 | 58.84 | 58.15 | 57.96 | 47.59 | 46.89 | 45.67 |
| MVAR | 78.81 | 49.27 | 53.60 | 72.08 | 36.78 | 40.08 | 63.92 | 24.57 | 25.17 | 62.91 | 62.85 | 62.34 | 58.42 | 58.39 | 57.36 | 51.61 | 51.16 | 49.73 |
| ERL-MVSC | 65.81 | 49.78 | 46.96 | 72.69 | 44.69 | 42.23 | 55.47 | 34.95 | 33.60 | 52.99 | 53.60 | 53.27 | 48.91 | 49.34 | 49.04 | 42.58 | 43.07 | 42.72 |
| AMSC | 63.73 | 22.51 | 22.60 | 65.04 | 24.34 | 23.53 | 64.79 | 26.02 | 26.52 | 65.15 | 63.95 | 62.48 | 60.33 | 59.54 | 58.84 | 52.65 | 51.93 | 50.93 |
| SLIM | 76.58 | 48.71 | 52.94 | 73.02 | 44.50 | 48.48 | 65.17 | 34.86 | 37.39 | 66.75 | 65.92 | 65.16 | 61.65 | 60.98 | 60.67 | 52.69 | 52.53 | 52.22 |
| **Ours** | **81.88** | **56.75** | **58.02** | **79.99** | **53.10** | **54.79** | **72.17** | **36.81** | **39.72** | **70.34** | **69.99** | **69.56** | **69.24** | **68.29** | **67.70** | **64.52** | **63.83** | **63.29** |

| Method | MNIST-USPS | | | | | | | | | AwA | | | | | | | | |
|---|---|---|---|---|---|---|---|---|---|---|---|---|---|---|---|---|---|---|
| AMMSS | 90.20 | 90.20 | 90.13 | 84.93 | 84.93 | 84.80 | 81.43 | 81.43 | 81.29 | 63.48 | 53.67 | 53.86 | 58.52 | 49.16 | 49.73 | 55.21 | 46.33 | 47.22 |
| AMGL | 93.54 | 93.54 | 93.50 | 89.61 | 89.61 | 89.53 | 85.75 | 85.75 | 85.61 | 57.74 | 49.25 | 48.78 | 52.94 | 44.82 | 44.07 | 47.45 | 40.09 | 39.47 |
| MLAN | 88.29 | 88.29 | 88.24 | 84.96 | 84.96 | 84.88 | 75.72 | 75.72 | 75.92 | 66.49 | 58.22 | 59.25 | 56.74 | 49.44 | 52.78 | 47.72 | 41.82 | 48.37 |
| AMUSE | 93.38 | 93.38 | 93.35 | 88.39 | 88.39 | 88.28 | 83.16 | 83.16 | 82.99 | 51.44 | 44.02 | 44.06 | 46.50 | 39.31 | 38.79 | 40.51 | 34.31 | 33.51 |
| FMSSL | 86.36 | 86.36 | 86.17 | 79.72 | 79.72 | 79.15 | 72.58 | 72.58 | 71.76 | 65.60 | 55.79 | 55.32 | 59.34 | 50.30 | 50.20 | 52.24 | 42.86 | 41.57 |
| FMSEL | 90.51 | 90.51 | 90.47 | 85.40 | 85.40 | 85.32 | 80.39 | 80.39 | 80.22 | 65.32 | 58.86 | 60.09 | 58.33 | 52.04 | 53.53 | 57.03 | 50.71 | 51.72 |
| CFMSC | 93.55 | 93.55 | 93.53 | 89.05 | 89.05 | 88.97 | 84.57 | 84.57 | 84.45 | 67.24 | 59.88 | 60.92 | 62.23 | 54.71 | 55.69 | 59.85 | 52.41 | 53.31 |
| MVAR | 79.65 | 79.65 | 79.54 | 71.73 | 71.73 | 71.40 | 66.04 | 66.04 | 66.08 | 69.07 | **61.87** | **62.91** | 63.57 | 56.99 | 58.23 | 59.15 | 52.35 | 54.92 |
| ERL-MVSC | 88.02 | 88.02 | 88.00 | 85.90 | 85.90 | 85.85 | 82.46 | 82.46 | 82.41 | 55.02 | 51.33 | 50.51 | 52.12 | 48.43 | 47.63 | 49.11 | 45.32 | 44.54 |
| AMSC | 88.72 | 88.72 | 88.56 | 83.16 | 83.16 | 83.00 | 83.83 | 83.83 | 83.60 | 64.34 | 54.74 | 54.06 | 55.90 | 45.76 | 44.83 | 52.94 | 43.29 | 42.52 |
| SLIM | 84.61 | 84.61 | 84.45 | 81.33 | 81.33 | 81.17 | 79.17 | 79.17 | 78.93 | 64.78 | 54.24 | 53.50 | 61.40 | 51.28 | 50.73 | 59.01 | 49.35 | 48.81 |
| **Ours** | **95.09** | **95.09** | **95.07** | **95.58** | **95.58** | **95.55** | **95.62** | **95.62** | **95.59** | **69.12** | 58.47 | 57.16 | **70.58** | 59.77 | 58.62 | **69.41** | 58.82 | 57.59 |

Table 3: Ablation study of **AGF-TI** under LAR is 5%.

| VMR=30% | CUB | | UCI-Digit | | Caltech101-20 | | OutScene | | MNIST-USPS | | Avg. | |
|---|---|---|---|---|---|---|---|---|---|---|---|---|
| | ACC | F1 | ACC | F1 | ACC | F1 | ACC | F1 | ACC | F1 | ACC | F1 |
| **AGF-TI** | **78.33** | **77.03** | **95.98** | **95.97** | **81.88** | **58.02** | **70.34** | **69.56** | **95.09** | **95.07** | **84.32** | **79.13** |
| w/o $\mathbf{T}_v$ | 72.12 | 70.61 | 88.48 | 88.34 | 42.71 | 21.61 | 27.35 | 26.91 | 82.01 | 81.70 | 62.53 | 57.83 |
| w/o $\alpha_v$ | 76.77 | 76.05 | 90.28 | 90.35 | 69.65 | 48.13 | 66.41 | 66.59 | 93.24 | 93.14 | 79.27 | 74.85 |
| w/o TI | 72.82 | 71.87 | 88.96 | 88.94 | 76.63 | 50.56 | 63.55 | 63.06 | 88.85 | 88.76 | 78.16 | 72.64 |
| VMR=70% | | | | | | | | | | | | |
| **AGF-TI** | **74.25** | **72.30** | **95.16** | **95.13** | **72.17** | **39.72** | **64.52** | **63.29** | **95.62** | **95.59** | **80.34** | **73.21** |
| w/o $\mathbf{T}_v$ | 53.40 | 51.19 | 55.58 | 54.56 | 38.54 | 8.29 | 30.62 | 27.11 | 78.12 | 77.61 | 51.25 | 43.75 |
| w/o $\alpha_v$ | 67.67 | 64.84 | 76.47 | 75.59 | 55.07 | 22.86 | 58.10 | 57.41 | 94.40 | 94.33 | 70.34 | 63.00 |
| w/o TI | 63.16 | 61.97 | 83.04 | 82.97 | 47.55 | 17.90 | 34.69 | 33.02 | 84.07 | 83.92 | 62.50 | 55.96 |

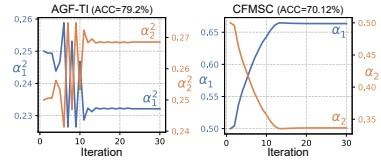

Figure 3: $\alpha$ comparison on CUB.

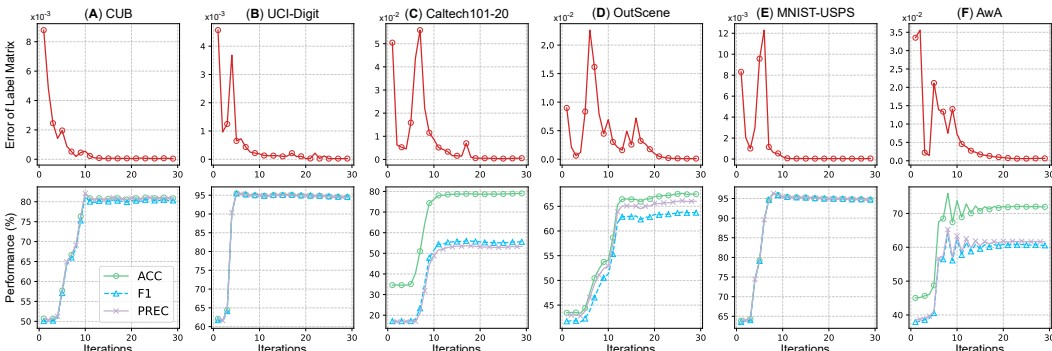

Figure 4: The iterative error and classification performance of **AGF-TI** during optimization process.

one could observe that (1) Each part plays a crucial role in performance improvement, suggesting the effectiveness of our **AGF-TI**. (2) When the number of views is large, *e.g.*, six for Caltech101-20 and four for OutScene, removing the permutation matrix results in a large performance degradation relative to the other datasets. (3) As VMR increased, the contribution of the tensorial imputation part is highlighted. For instance, removing the imputation part leads to an average performance drop of up to 17.8% when VRM is 70%, compared to a 6.2% degradation at 30% VRM in terms of ACC.

To further investigate the influence of the min-max scheme on graph fusion, we compare the evolution of the weight coefficients $\alpha$ from **AGF-TI** and a recent min-min scheme method, CFMSC. The comparison results on CUB under VMR=50% and LAR=5% are plotted in Fig. 3. Unlike the min-min framework, the proposed AGF operator alternatively magnifies the weight coefficient of each view by $\max_{\alpha}$ while alleviating the structural disparity across views by $\min_{\mathbf{P}}$. According to [38, 50], this alternative pattern could yield a robust structure against noisy perturbation by fully exploring complementary information from individual views, demonstrating the strength of AGF.

### 4.4 Model Analysis

In this section, we conduct model analyses under VMR=50% and LAR=5% conditions on the convergence behavior and parameter sensitivity *w.r.t.* $\beta_\lambda$ and $\rho$. More analyses on the number of anchors $m$, the trade-off parameter $\lambda$, and the computational costs of the TNN step are provided in Appendix F.3 and F.5.

**Convergence behavior.** To examine the convergence behavior of **AGF-TI**, we calculate the error value of the soft label matrix $\mathbf{F}$ and present its classification performance during iteration in Fig. 4. The error curves consistently exhibit early oscillations, subsequently undergoing a rapid decrease until convergence. The oscillation stage could be explained by the alternative pattern (illustrated in Fig. 3) discussed in Section 4.3. On the other hand, its performance substantially improves during this oscillation stage, further indicating the effectiveness of the proposed AGF.

**Parameter sensitivity.** **AGF-TI** introduces the regularized parameter $\beta_\lambda$ to control the connectivity of the fused graph and $\rho$ for tensorial imputation. We empirically analyze the impact of the two parameters on classification performance by tuning $\beta_\lambda$ and $\rho$ in the sets $\{2^0, 2^1, \ldots, 2^6\}$ and $\{10^0, \ldots, 10^6\}$, respectively. ACC results are recorded in Fig. 5 and more results are presented in Appendix F.3. Compared to $\rho$, $\beta_\lambda$ has great effects on **AGF-TI**, indicating that appropriate connectivity of the fused graph could further enhance classification performance. Besides, we find

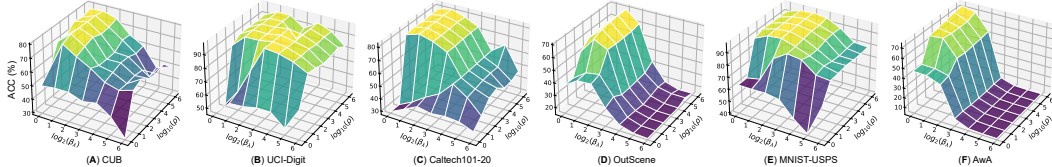

Figure 5: Parameter sensitivity analysis of $\beta_\lambda$ and $\rho$ in terms of Accuracy.

that **AGF-TI** exhibits stability with high performance within small ranges (*i.e.*, $\log_2(\beta_\lambda) \in [0, 2]$ and $\log_{10}(\rho) \in [2, 5]$) across various datasets, maintaining generalizability.

**Running time.** To empirically evaluate the complexity of the algorithms, we compare their running times on all datasets. For baselines requiring complete multi-view data, the total running time includes both the DMF completion process and its subsequent execution. The results are recorded in Table 4, and it can be observed that **AGF-TI** maintains an acceptable computational overhead. Although **AGF-TI** takes slightly longer than AMSC and SLIM due to additional tensor imputation and shared bipartite graph fusion, it consistently achieves advanced classification performance across all datasets. Moreover, thanks to the adopted anchor strategy, **AGF-TI**'s running time scales efficiently with the sample size, ensuring its practicality for large-scale applications.

Table 4: Running time (in seconds) of different methods under VMR=50% and LAR=5%.

| Method | CUB | UCI-Digit | Caltech101-20 | OutScene | MNIST-USPS | AwA |
|---|---|---|---|---|---|---|
| AMMSS | 4.31 | 7.80 | 86.25 | 15.61 | 48.58 | 621.34 |
| AMGL | 4.16 | 5.51 | 65.99 | 9.60 | 39.89 | 587.51 |
| MLAN | 4.68 | 20.29 | 141.65 | 74.99 | 150.06 | 1068.73 |
| AMUSE | 6.37 | 31.77 | 102.76 | 55.59 | 209.93 | 1361.76 |
| FMSSL | 4.77 | 5.81 | 82.94 | 24.51 | 74.75 | 808.80 |
| FMSEL | 6.45 | 32.70 | 120.18 | 60.12 | 234.50 | 1359.60 |
| CFMSC | 6.78 | 20.36 | 106.95 | 26.29 | 159.53 | 815.56 |
| MVAR | 4.26 | 5.40 | 69.84 | 10.36 | 39.30 | 601.27 |
| ERL-MVSC | 4.31 | 5.68 | 67.71 | 10.55 | 42.66 | 605.83 |
| AMSC | 0.85 | 2.15 | 11.03 | 4.40 | 3.73 | 33.54 |
| SLIM | 0.33 | 4.46 | 35.59 | 14.12 | 11.43 | 128.83 |
| **Ours** | 1.06 | 15.35 | 45.03 | 35.84 | 31.06 | 354.54 |

## 5  Conclusion

In this paper, we present the first study of the **S**ub-**C**luster **P**roblem (SCP) caused by missing views in graph-based multi-view semi-supervised learning. To address SCP, a novel **A**dversarial **G**raph **F**usion-based model with **T**ensorial **I**mputation (**AGF-TI**) is proposed. Our method benefits SCP in two aspects: (1) In AGF operator, the min-max optimization paradigm makes the model less sensitive to minor data fluctuations, enabling the ability to learn a robust fused graph against SCP. (2) The recovered local structures, imputed by high-order consistency information, further alleviate the impact of distorted graphs. For real-world applications, we adopt an anchor-based strategy to accelerate **AGF-TI**. An efficient iterative algorithm is developed to solve the proposed optimization problem. Extensive experimental results demonstrate the effectiveness of **AGF-TI**. This work focuses on the setting where the bipartite graphs are manually pre-constructed. Extending it to jointly optimize anchor selection and graph construction remains an interesting direction for future work.

## Acknowledgments

We thank Yansha Jia for her comments on this paper. This work was partially supported by the National Natural Science Foundation of China under Grant No. 62376281 and the Key NSF of China under Grant No. 62136005.

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

# A Basic Tensor Operators

We first summarize the notations used in this paper in Table 5.

Table 5: Summary of notations

| Notations | Annotation |
|---|---|
| $x, \mathbf{x}, \mathbf{X}, \mathcal{X}$ | scalar, vector, matrix, and tensor |
| $X_{ij}, \mathbf{X}(i,j)$ | the element in the $i$-th row and $j$-th column of $\mathbf{X}$ |
| $\mathbf{X}_{i\cdot}$ | the $i$-th row vector of $\mathbf{X}$ |
| $\mathcal{X}^k$ | the $k$-th frontal slice of tensor $\mathcal{X}$ |
| $\mathcal{X}_f = \text{fft}(\mathcal{X}, [], 3)$ | the fast Fourier transformation (FFT) |
| $n, V, m, \ell, c$ | the number of samples, views, anchors, labeled samples, and classes |
| $\mathbf{1}_n = [1, \ldots, 1]^\top$ | the all-ones column vector with $n$ elements |
| $\mathbf{I}_n \in \mathbb{R}^{n \times n}$ | the $n$ dimensional identity matrix |
| $\mathbf{X}_v \in \mathbb{R}^{n \times d_v}$ | the $d_v$ dimensional feature matrix of $v$-th view |
| $\mathbf{x}_i^{(v)} \in \mathbb{R}^{d_v \times 1}$ | the feature vector of $i$-th sample in $v$-th view |
| $\{y_i \in [c]\}_{i=1}^\ell$ | the label set of $\ell$ labeled samples |
| $\mathbf{Y} \in \{0, 1\}^{n \times c}$ | the one-hot label matrix of $n$ samples |
| $\mathbf{S}\ (\mathbf{S}_v) \in [0, 1]^{n \times n}$ | the symmetric similarity matrix among $n$ samples (of $v$-th view) |
| $\mathbf{D}_S$ | the diagonal degree matrix of $\mathbf{S}$ with entries $d_i = \sum_t \mathbf{S}_{it}$ |
| $\mathbf{L}_S = \mathbf{D}_S - \mathbf{S}$ | the Laplacian matrix of $\mathbf{S}$ |
| $\{\mathbf{a}_j^{(v)} \in \mathbb{R}^{d_v \times 1}\}_{j=1}^m$ | the set of $m$ anchors in $v$-th view |
| $\mathbf{Z}_v \in \Delta_n^m$ | the anchor-based bipartite matrix of $v$-th view |
| $\pi_v, \omega_v$ | the index sets of existing and missing samples in $v$-th view |
| $\mathbf{P} \in \Delta_n^m$ | the fused/consensus anchor-based bipartite graph |
| $\mathbf{T}_v \in \mathbb{R}^{m \times m}$ | the permutation matrix of $v$-th view |
| $\boldsymbol{\alpha} \in \Delta_V^1, \alpha_v \in [0, 1]$ | the weight coefficient vector and the weight coefficient of $v$-th view |
| $\mathbf{S}_P \in \mathbb{R}^{(n+m) \times (n+m)}$ | the similarity matrix among fused $n$ samples and $m$ anchors |
| $\hat{\mathbf{F}} = [\mathbf{F}; \mathbf{Q}] \in \mathbb{R}^{(n+m) \times c}$ | the soft label matrix of fused samples and anchors, where $\mathbf{F}$ is the soft label matrix of samples and $\mathbf{Q}$ is for anchors |
| $\hat{\mathbf{Y}} = [\mathbf{Y}; \mathbf{0}]$ | the one-hot true label matrix of samples and anchors |
| $\mathbf{B}, \hat{\mathbf{B}}$ | the diagonal regularization matrix |
| $\Phi(\cdot)$ | the merging and rotating operator |
| $\|\cdot\|_F, \|\cdot\|_2$ | the Frobenius norm and $\ell_2$ norm |
| $\|\cdot\|_\circledast$ | the t-SVD based tensor nuclear norm |
| $\Delta_n^m$ | $\{\zeta \in \mathbb{R}^{n \times m} \mid \zeta \mathbf{1}_m = \mathbf{1}_n, \zeta \geq 0\}$ |

Assume that $\mathcal{X} \in \mathbb{R}^{n_1 \times n_2 \times n_3}$ and $\mathcal{Y} \in \mathbb{R}^{n_2 \times n_4 \times n_3}$ are two third-order tensors. Then, we introduce some operators related to tensors.

• Transposition of tensor $\mathcal{X}^T \in \mathbb{R}^{n_2 \times n_1 \times n_3}$, which means that each frontal slice of the tensor is transposed.

• Cyclic expansion of the tensor $\text{circ}(\mathcal{X}) \in \mathbb{R}^{n_1 n_3 \times n_2 n_3}$:

$$\text{circ}(\mathcal{X}) = \begin{bmatrix} \mathcal{X}^1 & \mathcal{X}^{n_3} & \cdots & \mathcal{X}^2 \\ \mathcal{X}^2 & \mathcal{X}^1 & \cdots & \mathcal{X}^3 \\ \vdots & \vdots & \ddots & \vdots \\ \mathcal{X}^{n_3} & \mathcal{X}^{n_3-1} & \cdots & \mathcal{X}^1 \end{bmatrix}. \tag{18}$$

• Tensor unfolding and folding:

$$\text{unfold}(\mathcal{X}) = [\mathcal{X}^1, \mathcal{X}^2, \cdots, \mathcal{X}^{n_3}]^\top \in \mathbb{R}^{n_1 n_3 \times n_2}, \ \mathcal{X} = \text{fold}(\text{unfold}(\mathcal{X})). \tag{19}$$

• t-product $\mathcal{X} * \mathcal{Y} \in \mathbb{R}^{n_1 \times n_4 \times n_3}$:

$$\mathcal{X} * \mathcal{Y} = \text{fold}(\text{circ}(\mathcal{X}) \cdot \text{unfold}(\mathcal{Y})). \tag{20}$$

**Definition 2 (Orthogonal Tensor)** *The tensor $\mathcal{X}$ is orthogonal if $\mathcal{X}^\top * \mathcal{X} = \mathcal{X} * \mathcal{X}^\top = \mathcal{I}$, where $\mathcal{I} \in \mathbb{R}^{n_1 \times n_1 \times n_3}$ is an identity tensor whose first frontal slice is $\mathcal{I}^1 = \mathbf{I} \in \mathbb{R}^{n_1 \times n_1}$, and the other frontal slices are $\mathcal{I}^k = \mathbf{0}, \forall k = 2, 3, \ldots, n_3$.*

Based on basic tensor operations, tensor Singular Value Decomposition (t-SVD) is defined as follows.

**Definition 3 (t-SVD)** *Given a tensor $\mathcal{X} \in \mathbb{R}^{n_1 \times n_2 \times n_3}$, then the t-SVD of $\mathcal{X}$ is:*

$$\mathcal{X} = \mathcal{U} * \mathcal{S} * \mathcal{V}^\top, \tag{21}$$

*where $\mathcal{U} \in \mathbb{R}^{n_1 \times n_1 \times n_3}$ and $\mathcal{V} \in \mathbb{R}^{n_2 \times n_2 \times n_3}$ are orthogonal tensors, $\mathcal{S} \in \mathbb{R}^{n_1 \times n_2 \times n_3}$ is a f-diagonal tensor.*

# B  Optimization Details

## B.1  Update Formula of $\mathbf{Z}_{\omega_v}$

Recall that the subproblem of $\mathbf{Z}_{\omega_v}$ can be formulated as follows:

$$\arg \min_{\mathbf{Z}_{i\cdot}^{(v)}} \left\| \mathbf{Z}_{i\cdot}^{(v)} - \left( \mathbf{G}_{i\cdot}^{(v)} - \frac{1}{\eta} \left( \mathbf{W}_{i\cdot}^{(v)} - \lambda \alpha_v^2 (\mathbf{PT}_v)_{i\cdot} \right) \right) \right\|_2^2, \text{ s.t. } \mathbf{Z}_{i\cdot}^{(v)} \mathbf{1}_m = 1, \mathbf{Z}_{i\cdot}^{(v)} \geq 0, \ i \in \omega_v. \tag{22}$$

Denote $\mathbf{E}_{i\cdot}^{(v)} = \mathbf{W}_{i\cdot}^{(v)} - \lambda \alpha_v^2 (\mathbf{PT}_v)_{i\cdot}$, the Lagrangian function of problem (22) can be written as:

$$L(\mathbf{Z}_{i\cdot}^{(v)}, \zeta, \boldsymbol{\epsilon}_i) = \frac{1}{2} \left\| \mathbf{Z}_{i\cdot}^{(v)} - \left( \mathbf{G}_{i\cdot}^{(v)} - \frac{1}{\eta} \mathbf{E}_{i\cdot}^{(v)} \right) \right\|_2^2 - \zeta(\mathbf{Z}_{i\cdot}^{(v)} \mathbf{1}_m - 1) - \mathbf{Z}_{i\cdot}^{(v)} \boldsymbol{\epsilon}_i, \tag{23}$$

where $\zeta$ and $\boldsymbol{\epsilon}_i \geq 0$ are the Lagrange multipliers. The optimal solution $\mathbf{Z}_{i\cdot}^{(v)\star}$ should satisfy that the derivative of Eq. (23) *w.r.t.* $\mathbf{Z}_{i\cdot}^{(v)}$ is equal to zero, so we have

$$\mathbf{Z}_{i\cdot}^{(v)\star} - \left( \mathbf{G}_{i\cdot}^{(v)} - \frac{1}{\eta} \mathbf{E}_{i\cdot}^{(v)} \right) - \zeta \mathbf{1}_m - \boldsymbol{\epsilon}_i = \mathbf{0}. \tag{24}$$

We can rewrite it in the element-wise form:

$$Z_{ij}^{(v)\star} - \left( G_{ij}^{(v)} - \frac{1}{\eta} E_{ij}^{(v)} \right) - \zeta - \epsilon_{ij} = 0. \tag{25}$$

Note that $Z_{ij}^{(v)} \epsilon_{ij} = 0$ according to the KKT condition. Then, we have

$$Z_{ij}^{(v)\star} = \left( G_{ij}^{(v)} - \frac{1}{\eta} E_{ij}^{(v)} + \zeta \right)_+. \tag{26}$$

Each $\mathbf{Z}_{i\cdot}^{(v)}, i \in \omega_v$ can then be solved and we can update $\mathbf{Z}_{\omega_v}$.

## B.2  Optimization Algorithm of P and $\alpha$

After obtaining the reduced gradient $\nabla h(\boldsymbol{\alpha})$, we set the descent direction $\mathbf{g} = [g_1, g_2, \ldots, g_V]^\top$ with the following strategy:

$$g_v = \begin{cases} -[\nabla h(\boldsymbol{\alpha})]_u, & v = u, \\ -[\nabla h(\boldsymbol{\alpha})]_v, & \alpha_v > 0, v \neq u, \\ 0 & \alpha_v = 0, [\nabla h(\boldsymbol{\alpha})]_v > 0, \end{cases} \tag{27}$$

where $u$ is typically set as the index of the largest entry of $\boldsymbol{\alpha}$. Its main procedures of updating $\mathbf{P}$ and $\boldsymbol{\alpha}$ are summarized in Algorithm 1.

**Algorithm 1** Gradient Descent-based Optimization Algorithm for Updating $\boldsymbol{\alpha}$ and $\mathbf{P}$

---

**Input:** $\mathbf{P}, \{\mathbf{Z}_v, \mathbf{T}_v\}_{v=1}^V, \hat{\mathbf{F}}, \tilde{\mathbf{L}}_{S_P}, \lambda, \beta_\lambda$.
**Output:** Weight coefficient $\boldsymbol{\alpha}$ and fused graph $\mathbf{P}$.
1: **while** not converge **do**
2:     Calculate the soft label distance matrix $\mathbf{H}$ by $H_{ij} = \|\mathbf{F}_{i,.}/\sqrt{d_i} - \mathbf{Q}_{j,.}/\sqrt{d_{n+j}}\|_2^2$.
3:     Calculate the fused graph $\mathbf{P}$ by solving Eq. (11).
4:     Calculate the reduced gradient by Theorem 1 and Eq. (12).
5:     Calculate the descent gradient $\mathbf{g}$ by Eq. (27).
6:     Update weight coefficient $\boldsymbol{\alpha}_{t+1} \leftarrow \boldsymbol{\alpha}_t + \theta \mathbf{g}$ with the step length $\theta$.
7:     **if** $\max(|\boldsymbol{\alpha}_{t+1} - \boldsymbol{\alpha}_t| \leq 10^{-4})$ **then**
8:         Converge.
9:     **end if**
10: **end while**

---

### B.3  Update Formula of $\hat{\mathbf{F}}$

Recall that $\mathbf{S}_P = \begin{bmatrix} & \mathbf{P} \\ \mathbf{P}^\top & \end{bmatrix}$ and the normalized Laplacian matrix $\tilde{\mathbf{L}}_{S_P} = \mathbf{I}_{n+m} - \mathbf{D}_{S_P}^{-\frac{1}{2}} \mathbf{S}_P \mathbf{D}_{S_P}^{-\frac{1}{2}}$.

According to the definition of the degree matrix, $\mathbf{D}_{S_P}$ can be written as $\mathbf{D}_{S_P} = \begin{bmatrix} \mathbf{D}_r & \\ & \Lambda \end{bmatrix}$, where $\mathbf{D}_r \in \mathbb{R}^{n \times n}$ is diagonal matrix whose diagonal elements are row sums of $\mathbf{P}$ and $\Lambda \in \mathbb{R}^{m \times m}$ is a diagonal matrix whose diagonal elements are column sums of $\mathbf{P}$, *i.e.*, $\Lambda_{jj} = \sum_{i=1}^n P_{ij}$. Since $\mathbf{P}\mathbf{1}_m = \mathbf{1}_n$, we have $\mathbf{D}_r = \mathbf{I}_n$. Therefore, the normalized Laplacian matrix can be written as the following blockwise form,

$$\tilde{\mathbf{L}}_{S_P} = \mathbf{I}_{n+m} - \mathbf{D}_{S_P}^{-\frac{1}{2}} \mathbf{S}_P \mathbf{D}_{S_P}^{-\frac{1}{2}} = \begin{bmatrix} \mathbf{I}_n & -\mathbf{P}\Lambda^{-\frac{1}{2}} \\ -\Lambda^{-\frac{1}{2}}\mathbf{P}^\top & \mathbf{I}_m \end{bmatrix}. \tag{28}$$

### B.4  Update Formula of $\mathbf{T}_v$

The subproblem of $\mathbf{T}_v$ in Eq. (17) can be solved by the following theorem.

**Theorem 3** *Assume that $\mathbf{Z}_v^\top \mathbf{P} \in \mathbb{R}^{m \times m}$ in Eq. (17) has the singular value decomposition form as $\mathbf{Z}_v^\top \mathbf{P} = \mathbf{U}_v \boldsymbol{\Sigma}_v \mathbf{V}_v^\top$, where $\mathbf{U}_v, \boldsymbol{\Sigma}_v, \mathbf{V}_v \in \mathbb{R}^{m \times m}$. The optimization in Eq. (17) has a closed-form solution as follows,*

$$\mathbf{T}_v^\star = \mathbf{U}_v \mathbf{V}_v^\top. \tag{29}$$

*Proof.* Taking the equation $\mathbf{Z}_v^\top \mathbf{P} = \mathbf{U}_v \boldsymbol{\Sigma}_v \mathbf{V}_v^\top$, we can rewrite the Eq. (17) as,

$$\mathrm{Tr}(\mathbf{T}_v^\top \mathbf{U}_v \boldsymbol{\Sigma}_v \mathbf{V}_v^\top) = \mathrm{Tr}(\mathbf{V}_v^\top \mathbf{T}_v^\top \mathbf{U}_v \boldsymbol{\Sigma}_v). \tag{30}$$

Considering $\mathbf{E}_v = \mathbf{V}_v^\top \mathbf{T}_v^\top \mathbf{U}_v$, we have $\mathbf{E}_v \mathbf{E}_v^\top = \mathbf{V}_v^\top \mathbf{T}_v^\top \mathbf{U}_v \mathbf{U}_v^\top \mathbf{T}_v \mathbf{V}_v = \mathbf{I}$. Therefore, we can obtain:

$$\mathrm{Tr}(\mathbf{V}_v^\top \mathbf{T}_v^\top \mathbf{U}_v \boldsymbol{\Sigma}_v) = \mathrm{Tr}(\mathbf{E}_v \boldsymbol{\Sigma}_v) \leq \sum_{i=1}^m \sigma_i^{(v)}, \tag{31}$$

where $\sigma_i^{(v)}$ is the $i$-th diagonal element of $\boldsymbol{\Sigma}_v$. To maximize the value of Eq. (17), the solution should be given as $\mathbf{T}_v^\star = \mathbf{U}_v \mathbf{V}_v^\top$, thus achieving the maximum by satisfying the equality condition. $\qquad\square$

### B.5  Optimization Algorithm of AGF-TI

To solve the problem in Eq. (5), the algorithm of **AGF-TI** is summarized in Algorithm 2.

## C  Proof of Theorem 1

*Proof.* To prove Theorem 1, we first give a lemma:

---

**Algorithm 2** Optimization Algorithm of **AGF-TI**

---

**Input:** Anchor-based bipartite graphs $\{\mathbf{Z}_{\pi_v} \in \mathbb{R}^{|\pi_v| \times m}\}_{v=1}^V$, one-hot label matrix $\mathbf{Y}$, the number of categories $c$, the regularization parameters $\hat{\mathbf{B}}$, $\lambda$, $\beta$, and $\rho$.

**Output:** Labels of unlabeled samples $\tilde{\mathbf{Y}}_u$.

 1: For each $v \in [V]$, initialize $\mathbf{Z}_{\omega_v} = \frac{1}{m}\mathbf{1}_{|\omega_v|}\mathbf{1}_m^\top$, $\mathbf{T}_v = \mathbf{I}$, $\alpha_v = \frac{1}{V}$.
 2: Initialize auxiliary variable $\mathcal{G} = \mathbf{0}$ and Lagrange multipliers $\mathcal{W} = \mathbf{0}$, $\eta = 10^{-2}$, $\gamma_\eta = 2$, $\eta_{\max} = 10^{10}$.
 3: Calculate $\mathbf{P}$ by Eq. (11) with $\mathbf{H}$ is set to $\mathbf{0}$.
 4: Calculate $\mathbf{F}$ and $\mathbf{Q}$ according to Eq. (15).
 5: **while** not converge **do**
 6:     For each $v \in [V]$ and $i \in \omega_v$, update the $i$-th row of $\mathbf{Z}$ by solving Eq. (9).
 7:     Update weight coefficients $\boldsymbol{\alpha}$ and fused graph $\mathbf{P}$ according to Algorithm 1.
 8:     Update $\mathbf{F}$ and $\mathbf{Q}$ by Eq. (15).
 9:     Update $\mathcal{G}$ by Theorem 2.
10:     For each $v \in [V]$, update $\mathbf{T}_v$ by solving Eq. (17).
11:     Update $\mathcal{W} = \mathcal{W} + \eta(\mathcal{Z} - \mathcal{G})$.
12:     Update penalty parameter $\eta = \min(\gamma_\eta \eta, \eta_{\max})$.
13: **end while**
14: **return** classification results: $\tilde{\mathbf{Y}}_u \triangleq \{\tilde{y}_i | \tilde{y}_i = \arg\max_{j \in [c]} F_{ij}\}_{i=\ell+1}^n$.

---

**Lemma 1** [35] *Given a function $g(\mathbf{x}, \mathbf{u})$ where $\mathbf{x}$ and $\mathbf{u}$ belongs to compact normed spaces $\mathcal{X}$ and $\mathcal{U}$, respectively. Assume that $g(\mathbf{x}, \cdot)$ is differentiable over $\mathcal{X}$, $g(\mathbf{x}, \mathbf{u})$ and $\frac{\partial g(\mathbf{x}, \mathbf{u})}{\partial \mathbf{u}}$ are continuous on $\mathcal{X} \times \mathcal{U}$, then the optimal value function $h(\mathbf{u}) \triangleq Sup_{\mathbf{x} \in \mathcal{X}} \, g(\mathbf{x}, \mathbf{u})$ is differentiable and $\frac{\partial h(\mathbf{u}_0)}{\partial \mathbf{u}} = \frac{\partial g(\mathbf{x}^\star, \mathbf{u}_0)}{\partial \mathbf{u}}$ at point $\mathbf{u}_0$ if $g(\mathbf{x}, \mathbf{u}_0)$ has a unique maximizer $\mathbf{x}^\star$.*

Then, we construct the following function:

$$g(\mathbf{P}, \boldsymbol{\alpha}) = \lambda \sum_v \alpha_v^2 \mathrm{Tr}(\mathbf{P}^\top (\mathbf{Z}_v \mathbf{T}_v)) - \beta_\lambda \|\mathbf{P}\|_F^2 - \mathrm{Tr}(\hat{\mathbf{F}}^\top \mathbf{L}_{\hat{\mathbf{S}}_P} \hat{\mathbf{F}}), \tag{32}$$

where $\mathbf{P} \in \Delta_n^m$ and $\boldsymbol{\alpha} \in \Delta_V^1$. Based on the property of the normalized Laplacian matrix, the relationship $\mathrm{Tr}(\mathbf{F}^\top \mathbf{L}_{\hat{\mathbf{S}}_P} \hat{\mathbf{F}}) = \sum_i^n \sum_j^m \|\mathbf{F}_{i,\cdot}/\sqrt{d_i} - \mathbf{Q}_{j,\cdot}/\sqrt{d_{n+j}}\|_2^2 P_{ij}$ holds. Evidently, $g(\mathbf{P}, \cdot)$ is differentiable over $\Delta_n^m$, and $g(\mathbf{P}, \boldsymbol{\alpha})$ and $\frac{\partial g(\mathbf{P}, \boldsymbol{\alpha})}{\partial \boldsymbol{\alpha}}$ are continuous. Recall that $\Delta_n^m = \{\zeta \in \mathbb{R}^{n \times m} | \zeta \mathbf{1}_m = \mathbf{1}_n, \zeta \geq 0\}$ is a compact space. Considering the optimal value function of $g(\mathbf{P}, \boldsymbol{\alpha})$ w.r.t. $\boldsymbol{\alpha}$, we have:

$$\mathrm{Sup}_{\mathbf{P} \in \Delta_n^m} g(\mathbf{P}, \boldsymbol{\alpha}) = \max_{\mathbf{P} \in \Delta_n^m} \lambda \sum_v \alpha_v^2 \mathrm{Tr}(\mathbf{P}^\top (\mathbf{Z}_v \mathbf{T}_v)) - \beta_\lambda \|\mathbf{P}\|_F^2 - \mathrm{Tr}(\hat{\mathbf{F}}^\top \mathbf{L}_{\hat{\mathbf{S}}_P} \hat{\mathbf{F}}), \tag{33}$$

which has the exact form as $h(\boldsymbol{\alpha})$ in Eq. (10). According to Lemma 1, the differentiable property of Eq. (33), *i.e.*, $h(\boldsymbol{\alpha})$, at a given point $\boldsymbol{\alpha}^0$ depends on whether we can find an unique maximizer $\mathbf{P}^\star$ for the inner optimization problem $\max_{\mathbf{P} \in \Delta_n^m} g(\mathbf{P}, \boldsymbol{\alpha}^0)$, *i.e.*, problem in Eq. (11). Since the feasible region $\Delta_n^m$ is a closed convex set and $g(\mathbf{P}, \boldsymbol{\alpha}^0)$ is strictly convex, the Hilbert projection theorem guarantees the uniqueness of the maximizer for any given $\boldsymbol{\alpha}$. Following that, we can conclude that $h(\boldsymbol{\alpha})$ is differentiable and its gradient can be calculated by:

$$\frac{\partial h(\boldsymbol{\alpha})}{\partial \alpha_v} = 2\lambda \alpha_v \mathrm{Tr}(\mathbf{P}^{\star\top} \mathbf{Z}_v \mathbf{T}_v), \tag{34}$$

where $\mathbf{P}^\star$ is the optimal solution of the inner optimization problem. $\qquad \square$

## D   Theoretical Analysis

### D.1   Convergence Analysis

In essence, Algorithm 2 iterates between the following two steps until convergence:

(1) **Inner Step:** Solve the min-max optimization problem in Eq. (10) for fixed variables $(\{\mathbf{Z}_{\omega_v}, \mathbf{T}_v\}_v^V, \hat{\mathbf{F}}, \mathcal{G}, \mathcal{W})$ using a reduced gradient descent method, *i.e.*, Algorithm 1, to update inner variables $\boldsymbol{\alpha}$ and $\mathbf{P}$.

(2) **Outer Step:** Update other outer variables through ADMM with augmented Lagrangian function, *i.e.*, $\mathcal{J}(\{\mathbf{Z}_{\omega_v}, \mathbf{T}_v\}_v^V, \hat{\mathbf{F}}, \boldsymbol{\alpha}^\star, \mathbf{P}^\star, \mathcal{G}, \mathcal{W})$, where $\boldsymbol{\alpha}^\star$ and $\mathbf{P}^\star$ denote the optimal inner variables for given outer variables.

To prove the convergence of Algorithm 2, we proceed in three parts. First, we prove that the solution of the **Inner Step** obtained by Algorithm 1 is the global optimum. Second, leveraging a mild assumption on the coupling between inner and outer variables, we construct a Lyapunov function based on an idealized augmented Lagrangian to show that the ADMM procedure of the **Outer Step** produces a bounded sequence $\{\{\mathbf{Z}_{\omega_v}^{(k)}, \mathbf{T}_v^{(k)}\}_v^V, \hat{\mathbf{F}}^{(k)}, \mathcal{G}^{(k)}, \mathcal{W}^{(k)}\}_{k=1}^\infty$. Third, we prove that Algorithm 2 converges to a stationary Karush-Kuhn-Tucker (KKT) point of the original max-min-max problem in Eq. (5).

- **Part I: Proof of the global convergence of Algorithm 1 used in Inner Step**

**Theorem 4** $h(\boldsymbol{\alpha})$ *in Eq. (10) is convex* w.r.t. $\boldsymbol{\alpha}$.

*Proof.* For any $\boldsymbol{\alpha}_1$ and $\boldsymbol{\alpha}_2 \in \Delta_V^1$, and $0 < \gamma < 1$, the following form holds:

$h(\gamma\boldsymbol{\alpha}_1 + (1-\gamma)\boldsymbol{\alpha}_2)$

$$
\begin{aligned}
&= \max_{\mathbf{P}\in\Delta_n^m} \lambda\mathrm{Tr}\left(\sum_v (\gamma\alpha_{1v} + (1-\gamma)\alpha_{2v})^2 \mathbf{P}^\top(\mathbf{Z}_v\mathbf{T}_v)\right) - (\gamma+1-\gamma)\overbrace{\left(\beta_\lambda\|\mathbf{P}\|_F^2 - \mathrm{Tr}(\hat{\mathbf{F}}^\top\tilde{\mathbf{L}}_{S_P}\hat{\mathbf{F}})\right)}^{C} \\
&\le \max_{\mathbf{P}\in\Delta_n^m} \lambda\mathrm{Tr}\left(\sum_v (\gamma\alpha_{1v}^2 + (1-\gamma)\alpha_{2v}^2)\mathbf{P}^\top(\mathbf{Z}_v\mathbf{T}_v)\right) - (\gamma+1-\gamma)C \\
&\le \gamma\left(\max_{\mathbf{P}\in\Delta_n^m} \lambda\mathrm{Tr}\left(\sum_v \alpha_{1v}^2\mathbf{P}^\top(\mathbf{Z}_v\mathbf{T}_v)\right) - C\right) + (1-\gamma)\left(\max_{\mathbf{P}\in\Delta_n^m} \lambda\mathrm{Tr}\left(\sum_v \alpha_{2v}^2\mathbf{P}^\top(\mathbf{Z}_v\mathbf{T}_v)\right) - C\right) \\
&= \gamma h(\boldsymbol{\alpha}_1) + (1-\gamma)h(\boldsymbol{\alpha}_2).
\end{aligned}
\tag{35}
$$

Eq. (35) validates that $h(\boldsymbol{\alpha})$ satisfies the definition of convex function. $\square$

Algorithm 1 conducts the reduced gradient descent on a continuously differentiable function $h(\boldsymbol{\alpha})$, which is defined on the simplex $\{\boldsymbol{\alpha}\in\mathbb{R}^{V\times1} | \sum_{v=1}^V \alpha_v = 1, \alpha_v \ge 0, \forall v\}$. Hence, it converges to the minimum of $h(\boldsymbol{\alpha})$. According to Theorem 4, we have the following corollary.

**Corollary 1** *The solution of the **Inner Step** obtained by Algorithm 1 is the global optimum.*

- **Part II: Proof of the boundedness of the sequence generated by ADMM in Outer Step**

Since the **Outer Step** is based on the fixed inner variables $\boldsymbol{\alpha}$ and $\mathbf{P}$, the following mild assumption allows us to use the idealized augmented Lagrangian function, simplifying the convergence analysis.

**Assumption 1 (Sufficient Inner Optimization)** *At each iteration $k$, the inner optimization finds a sufficiently accurate solution $\boldsymbol{\alpha}^{(k)}$ and $\mathbf{P}^{(k)}$ such that for some $\epsilon > 0$:*

$$
\|\boldsymbol{\alpha}^{(k)} - \boldsymbol{\alpha}^\star\|_2 + \|\mathbf{P}^{(k)} - \mathbf{P}^\star\|_F \le \epsilon
\tag{36}
$$

*where $(\boldsymbol{\alpha}^\star, \mathbf{P}^\star)$ is the exact saddle point of the min-max problem in Eq. (10), given other variables.*

According to Corollary 1 and the convergence condition adopted in Algorithm 1, this assumption is satisfiable. Besides, due to the used $\ell_2$ norm and Frobenius norm, the augmented Lagrangian function in Eq. (6) is strongly-convex in $\boldsymbol{\alpha}$ and strongly-concave in $\mathbf{P}$. This favorable property ensures that the optimal solution of **Inner Step** is Lipschitz continuous *w.r.t.* the outer variables. Therefore, in **Outer Step**, the augmented Lagrangian function in Eq. (6) at each iteration can be idealized as:

$$
\begin{aligned}
\tilde{\mathcal{J}}(\{\mathbf{Z}_{\omega_v}^{(k)}, \mathbf{T}_v^{(k)}\}_v^V, \hat{\mathbf{F}}^{(k)}, \mathcal{G}^{(k)}, \mathcal{W}^{(k)}) &\triangleq \min_{\boldsymbol{\alpha}\in\Delta_V^1} \max_{\mathbf{P}\in\Delta_n^m} \mathcal{J}(\{\mathbf{Z}_{\omega_v}^{(k)}, \mathbf{T}_v^{(k)}\}_v^V, \hat{\mathbf{F}}^{(k)}, \boldsymbol{\alpha}, \mathbf{P}, \mathcal{G}^{(k)}, \mathcal{W}^{(k)}) \\
&\approx \mathcal{J}(\{\mathbf{Z}_{\omega_v}^{(k)}, \mathbf{T}_v^{(k)}\}_v^V, \hat{\mathbf{F}}^{(k)}, \boldsymbol{\alpha}^{(k)}, \mathbf{P}^{(k)}, \mathcal{G}^{(k)}, \mathcal{W}^{(k)}).
\end{aligned}
\tag{37}
$$

Following [29], we can prove the following theorem.

**Theorem 5** *If Assumption 1 holds, the sequence $\{\{\mathbf{Z}_{\omega_v}^{(k)}, \mathbf{T}_v^{(k)}\}_v^V, \hat{\mathbf{F}}^{(k)}, \mathcal{G}^{(k)}, \mathcal{W}^{(k)}\}_{k=1}^\infty$ generated by the ADMM procedure of the **Outer Step** is bounded.*

*Proof.* We first introduce a lemma.

**Lemma 2** [51] *Suppose that* $F : \mathbb{R}^{m \times n} \mapsto \mathbb{R}$ *is defined as* $F(\mathbf{X}) = f \circ \boldsymbol{\sigma}(\mathbf{X}) = f(\sigma_1(\mathbf{X}), \dots, \sigma_r(\mathbf{X}))$, *where* $\mathbf{X} = \mathbf{U} \cdot Diag(\boldsymbol{\sigma}(\mathbf{X})) \cdot \mathbf{V}^\top$ *is the normal SVD of matrix* $\mathbf{X} \in \mathbb{R}^{m \times n}$, $r = \min(m, n)$, *and* $f(\cdot) : \mathbb{R}^r \mapsto \mathbb{R}$ *be differentiable and absolutely symmetric at* $\boldsymbol{\sigma}(\mathbf{X})$. *Then the subdifferential of* $F(\mathbf{X})$ *at* $\mathbf{X}$ *is*

$$\frac{\partial F(\mathbf{X})}{\partial \mathbf{X}} = \mathbf{U} \cdot Diag(\partial f(\boldsymbol{\sigma}(\mathbf{X}))) \cdot \mathbf{V}^\top, \tag{38}$$

*where* $\partial f(\boldsymbol{\sigma}(\mathbf{X})) = \left( \frac{\partial f(\sigma_1(\mathbf{X}))}{\partial \mathbf{X}}, \dots, \frac{\partial f(\sigma_r(\mathbf{X}))}{\partial \mathbf{X}} \right)$.

Then, we construct the following Lyapunov function to prove the outer sequence $\{\{\mathbf{Z}_{\omega_v}^{(k)}, \mathbf{T}_v^{(k)}\}_v^V, \hat{\mathbf{F}}^{(k)}, \mathcal{G}^{(k)}, \mathcal{W}^{(k)}\}_{k=1}^\infty$ is bounded.

$$\mathcal{V}_{\eta_k}(\{\mathbf{Z}_{\omega_v}^{(k)}, \mathbf{T}_v^{(k)}\}_v^V, \hat{\mathbf{F}}^{(k)}, \mathcal{G}^{(k)}, \mathcal{W}^{(k)}) = -\tilde{\mathcal{J}}(\{\mathbf{Z}_{\omega_v}^{(k)}, \mathbf{T}_v^{(k)}\}_v^V, \hat{\mathbf{F}}^{(k)}, \mathcal{G}^{(k)}, \mathcal{W}^{(k)}) + \|\hat{\mathbf{Y}}\|_F^2.$$

$$= \text{Tr}\left( \hat{\mathbf{F}}^{(k)\top} \tilde{\mathbf{L}}_{S_{P^\star}} \hat{\mathbf{F}}^{(k)} \right) + \text{Tr}\left( (\hat{\mathbf{F}}^{(k)} - \hat{\mathbf{Y}})^\top \hat{\mathbf{B}} (\hat{\mathbf{F}}^{(k)} - \hat{\mathbf{Y}}) \right) - \lambda \sum_v \alpha_v^{\star 2} \text{Tr}\left( \mathbf{P}^{\star\top} (\mathbf{Z}_v^{(k)} \mathbf{T}_v^{(k)}) \right)$$

$$+ \rho \|\mathcal{G}^{(k)}\|_\circledast + \langle \mathcal{W}^{(k)}, \mathcal{Z}^{(k)} - \mathcal{G}^{(k)} \rangle + \frac{\eta_k}{2} \|\mathcal{Z}^{(k)} - \mathcal{G}^{(k)}\|_F^2, \tag{39}$$

where $\|\hat{\mathbf{Y}}\|_F^2$ is a constant. To prove the multiplier sequence $\{\mathcal{W}^{(k)}\}_{k=1}^\infty$ is bounded. We derive the first-order optimality condition of $\mathcal{G}$ in the updating rule:

$$\partial \|\mathcal{G}^{(k+1)}\|_\circledast = \mathcal{W}^{(k)} + \eta_k (\mathcal{Z}^{(k+1)} - \mathcal{G}^{(k+1)}) = \mathcal{W}^{(k+1)}. \tag{40}$$

Let $\mathcal{U} * \mathcal{S} * \mathcal{V}^\top$ be the t-SVD of tensor $\mathcal{G}$. According to Lemma 2 and Definition 1, we have

$$\left\| \partial \|\mathcal{G}^{(k+1)}\|_\circledast \right\|_F^2 = \left\| \frac{1}{n} \mathcal{U} * \text{ifft}(\partial \mathcal{S}_f, [], 3) * \mathcal{V}^\top \right\|_F^2$$

$$= \frac{1}{n^3} \|\partial \mathcal{S}_f\|_F^2 \le \frac{1}{n^3} \sum_{i=1}^n \sum_{j=1}^{\min(m, V)} 1, \tag{41}$$

which implies $\partial \|\mathcal{G}^{(k+1)}\|_\circledast$ is bounded. From Eq. (40), we can infer that the sequence $\{\mathcal{W}^{(k)}\}_{k=1}^\infty$ is also bounded.

Note that all subproblems of the ADMM procedure in the **Outer Step**, *i.e.*, Eq. (9), Eq. (13), Eq. (16), and Eq. (17), are closed, proper and convex, we can infer that

$$\mathcal{V}_{\eta_k}(\{\mathbf{Z}_{\omega_v}^{(k+1)}, \mathbf{T}_v^{(k+1)}\}_v^V, \hat{\mathbf{F}}^{(k+1)}, \mathcal{G}^{(k+1)}, \mathcal{W}^{(k)}) \le \mathcal{V}_{\eta_k}(\{\mathbf{Z}_{\omega_v}^{(k)}, \mathbf{T}_v^{(k)}\}_v^V, \hat{\mathbf{F}}^{(k)}, \mathcal{G}^{(k)}, \mathcal{W}^{(k)})$$

$$= \mathcal{V}_{\eta_{k-1}}(\{\mathbf{Z}_{\omega_v}^{(k)}, \mathbf{T}_v^{(k)}\}_v^V, \hat{\mathbf{F}}^{(k)}, \mathcal{G}^{(k)}, \mathcal{W}^{(k-1)}) + \frac{\eta_k + \eta_{k-1}}{2\eta_{k-1}^2} \|\mathcal{W}^{(k)} - \mathcal{W}^{(k-1)}\|_F^2. \tag{42}$$

By summing the right-hand side of Eq. (42) from $k = 1$ to $n$, we have

$$\mathcal{V}_{\eta_k}(\{\mathbf{Z}_{\omega_v}^{(k+1)}, \mathbf{T}_v^{(k+1)}\}_v^V, \hat{\mathbf{F}}^{(k+1)}, \mathcal{G}^{(k+1)}, \mathcal{W}^{(k)}) \le \mathcal{V}_{\eta_0}(\{\mathbf{Z}_{\omega_v}^1, \mathbf{T}_v^1\}_v^V, \hat{\mathbf{F}}^1, \mathcal{G}^1, \mathcal{W}^0)$$

$$+ \sum_{k=1}^n \frac{\eta_k + \eta_{k-1}}{2\eta_{k-1}^2} \|\mathcal{W}^{(k)} - \mathcal{W}^{(k-1)}\|_F^2. \tag{43}$$

Since $\sum_{k=1}^n \frac{\eta_k + \eta_{k-1}}{2\eta_{k-1}^2} < \infty$, $\mathcal{V}_{\eta_0}(\{\mathbf{Z}_{\omega_v}^1, \mathbf{T}_v^1\}_v^V, \hat{\mathbf{F}}^1, \mathcal{G}^1, \mathcal{W}^0)$ is finite, and the sequence $\{\mathcal{W}^{(k)}\}_{k=1}^\infty$ is bounded, we can derive $\mathcal{V}_{\eta_k}(\{\mathbf{Z}_{\omega_v}^{(k+1)}, \mathbf{T}_v^{(k+1)}\}_v^V, \hat{\mathbf{F}}^{(k+1)}, \mathcal{G}^{(k+1)}, \mathcal{W}^{(k)})$ is bounded. Recall the specific Lyapunov function in Eq. (39), except $-\alpha_v^{\star 2} \text{Tr}\left( \mathbf{P}^{\star\top} (\mathbf{Z}_v^{(k+1)} \mathbf{T}_v^{(k+1)}) \right)$, the other terms are nonnegative. Considering the constrains $\mathbf{P}, \mathbf{Z}_v \in \Delta_n^m, \boldsymbol{\alpha} \in \Delta_V^1, \mathbf{T}_v^\top \mathbf{T}_v = \mathbf{I}_n (\forall v \in [V])$, we can infer $\alpha_v^{\star 2} \text{Tr}\left( \mathbf{P}^{\star\top} (\mathbf{Z}_v^{(k+1)} \mathbf{T}_v^{(k+1)}) \right) < \infty$ holds. Thus, we can deduce that each term of Eq. (39) is bounded.

The boundedness of $\|\mathcal{G}^{(k+1)}\|_\circledast$ suggests that all singular values of $\mathcal{G}^{(k+1)}$ are bounded. Based on the following equation

$$\|\mathcal{G}^{(k+1)}\|_F^2 = \frac{1}{n} \|\mathcal{G}_f^{(k+1)}\|_F^2 = \frac{1}{n} \sum_{i=1}^n \sum_{j=1}^{\min(m, V)} (\mathcal{S}_f^i(j, j))^2, \tag{44}$$

the boundedness of the sequence $\{\mathcal{G}^{(k)}\}_{k=1}^{\infty}$ is ensured.

Since $\hat{\mathbf{B}}$ and $\hat{\mathbf{Y}}$ are constants, the boundedness of $\mathrm{Tr}((\hat{\mathbf{F}}^{(k+1)} - \hat{\mathbf{Y}})^{\top}\hat{\mathbf{B}}(\hat{\mathbf{F}}^{(k+1)} - \hat{\mathbf{Y}}))$ guarantees the boundedness of the sequence $\{\hat{\mathbf{F}}^{(k+1)}\}_{k=1}^{\infty}$.

Furthermore, according to the update formulas in Eq. (26) and Eq. (17), it could be deduced that these sequences $\{\mathbf{Z}_{\omega_v}^{(k)}\}_{k=1}^{\infty}$ and $\{\mathbf{T}_v^{(k)}\}_{k=1}^{\infty}$ are bounded. $\qquad\square$

• **Part III: Proof of the convergence of Algorithm 2**

Leveraging Corollary 1 and Theorem 5, the convergence of Algorithm 2 is guaranteed by the following theorem.

**Theorem 6** *If Assumption 1 holds, Algorithm 2 will converge to a stationary point of the problem in Eq. (5).*

*Proof.* Assume the sequence $\{\{\mathbf{Z}_{\omega_v}^{(k)}, \mathbf{T}_v^{(k)}\}_v^V, \hat{\mathbf{F}}^{(k)}, \boldsymbol{\alpha}^{(k)}, \mathbf{P}^{(k)}, \mathcal{G}^{(k)}, \mathcal{W}^{(k)}\}_{k=1}^{\infty}$ is generated by Algorithm 2 during iteration. The boundedness of $\{\{\mathbf{Z}_{\omega_v}^{(k)}, \mathbf{T}_v^{(k)}\}_v^V, \hat{\mathbf{F}}^{(k)}, \mathcal{G}^{(k)}, \mathcal{W}^{(k)}\}_{k=1}^{\infty}$ is guaranteed by Theorem 5. Since the feasible region of variables $\mathbf{P}$ and $\boldsymbol{\alpha}$ are both closed convex sets, the boundedness of $\{\boldsymbol{\alpha}^{(k)}, \mathbf{P}^{(k)}\}_{k=1}^{\infty}$ naturally holds.

According to the Weierstrass-Bolzano theorem [52], there is at least one accumulation point of the sequence $\{\{\mathbf{Z}_{\omega_v}^{(k)}, \mathbf{T}_v^{(k)}\}_v^V, \hat{\mathbf{F}}^{(k)}, \boldsymbol{\alpha}^{(k)}, \mathbf{P}^{(k)}, \mathcal{G}^{(k)}, \mathcal{W}^{(k)}\}_{k=1}^{\infty}$, we assume the one of the points as $(\{\mathbf{Z}_{\omega_v}^{\infty}, \mathbf{T}_v^{\infty}\}_v^V, \hat{\mathbf{F}}^{\infty}, \boldsymbol{\alpha}^{\infty}, \mathbf{P}^{\infty}, \mathcal{G}^{\infty}, \mathcal{W}^{\infty})$, *i.e.*,

$$\lim_{k\to\infty}(\{\mathbf{Z}_{\omega_v}^{(k)}, \mathbf{T}_v^{(k)}\}_v^V, \hat{\mathbf{F}}^{(k)}, \boldsymbol{\alpha}^{(k)}, \mathbf{P}^{(k)}, \mathcal{G}^{(k)}, \mathcal{W}^{(k)}) = (\{\mathbf{Z}_{\omega_v}^{\infty}, \mathbf{T}_v^{\infty}\}_v^V, \hat{\mathbf{F}}^{\infty}, \boldsymbol{\alpha}^{\infty}, \mathbf{P}^{\infty}, \mathcal{G}^{\infty}, \mathcal{W}^{\infty}).$$
(45)

From the update formula of $\mathcal{W}$, we have

$$\mathcal{Z}^{(k+1)} - \mathcal{G}^{(k+1)} = (\mathcal{W}^{(k+1)} - \mathcal{W}^{(k)})/\eta^{(k)}.$$
(46)

Leveraging the boundedness of $\{\mathcal{W}^{(k)}\}_{k=1}^{\infty}$, we can obtain $\mathcal{Z}^{\infty} - \mathcal{G}^{\infty} = 0$.

Due to the first-order optimality condition of $\mathcal{G}^{(k)}$, we can derive $\mathcal{W}^{\infty} \in \partial\|\mathcal{G}^{\infty}\|_{\circledast}$. Considering the closed-form solutions in Eq. (26), Eq. (15), Eq. (17), and Theorem 2, $(\{\mathbf{Z}_{\omega_v}^{\infty}, \mathbf{T}_v^{\infty}\}_v^V, \hat{\mathbf{F}}^{\infty}, \mathcal{G}^{\infty})$ satisfies their corresponding KKT conditions of the problem in Eq. (5). Furthermore, the global convergence property in Corollary 1 ensures that $\boldsymbol{\alpha}^{\infty}$ and $\mathbf{P}^{\infty}$ also satisfy the KKT condition.

Thus, the accumulation point $(\{\mathbf{Z}_{\omega_v}^{\infty}, \mathbf{T}_v^{\infty}\}_v^V, \hat{\mathbf{F}}^{\infty}, \boldsymbol{\alpha}^{\infty}, \mathbf{P}^{\infty}, \mathcal{G}^{\infty}, \mathcal{W}^{\infty})$ generated by Algorithm 2 satisfied the KKT condition of the problem in Eq. (5). $\qquad\square$

### D.2 Time and Space Complexity Analysis

**Time complexity analysis.** For the proposed **AGF-TI**, the main time complexity is focused on solving for the variables $\mathbf{Z}_{\omega_v}, \mathbf{P}, \boldsymbol{\alpha}, \hat{\mathbf{F}}, \mathcal{G}$, and $\mathbf{T}_v$. The update of the variables $\{\mathbf{Z}_{\omega_v}\}_{v=1}^V$ requires $\mathcal{O}(\sum_v^V |\omega_v|(m^2 + m))$. In Algorithm 1, calculating $\mathbf{H}$, $\mathbf{P}$, $\mathbf{g}$, and the optimal step for each update needs $\mathcal{O}(nm(c+1))$, $\mathcal{O}(nm(mV+1))$, $\mathcal{O}(nm^2V)$, and $\mathcal{O}(Vt_1)$ computational complexity, respectively, where $t_1$ is the number of iterations. Thus, the time complexity of updating $\mathbf{P}$ and $\boldsymbol{\alpha}$ is $\mathcal{O}(nm(2mV+c+2) + Vt_1)$. For $\hat{\mathbf{F}}$, it requires $\mathcal{O}(nm^2)$ for each iteration. For tensor $\mathcal{G}$, each update needs fast Fourier transformation (FFT), inverse FFT, and t-SVD operations, corresponding to a computational complexity of $\mathcal{O}(nmV\log(nV) + nmV^2)$. The update of $\{\mathbf{T}_v\}_{v=1}^V$ requires $\mathcal{O}(nm^2 + 2m^3)$ for each iteration. Assume $t_2$ iterations are required to achieve convergence. Considering $m, c, V, t_1 \ll n$, the overall time complexity of the optimization phase is $\mathcal{O}(t_2(nmV\log(nV) + nm^2V + nmV^2))$.

**Space complexity analysis.** For our proposed method, the major memory costs are various anchored matrices and tensors. According to the optimization strategy in §3.2, the space complexity of **AGF-TI** is $\mathcal{O}(n(mV + c))$.

# E  Experimental Details

## E.1  Detailed Description of Datasets

- *CUB*[2] includes 11,788 samples of 200 bird species. Following [41], we select the first 10 bird species with 1,024-d deep visual features from GoogLeNet and 300-d text features using the doc2vec model.

- *UCI-Digit*[3] contains 2,000 images for 0 to 9 ten digit classes, and each class has 200 data points. Following [42], we use 76-d Fourier coefficients of the character shapes, 216-d profile correlations, and 64-d Karhunen-Love coefficients as three views.

- *Caltech101-20*[4] is an image dataset for object recognition tasks, which includes 2,386 images of 20 classes. For each image, six features are extracted: 48-d Gabor, 40-d Wavelet Moments, 254-d CENTRIST, 1,984-d histogram of oriented gradient (HOG), 512-d GIST, and 928-d local binary patterns (LBP).

- *OutScene*[5] consists of 2,688 images belonging to 8 outdoor scene categories. Following [43], we use the same four features: 432-d Color, 512-d GIST, 256-d HOG, and 48-d LBP.

- *MNIST-USPS* [44] comprises 5,000 samples distributed over 10 digits, and the 784-d MNIST image and the 256-d USPS image are used as two views.

- *AwA*[6] (Animal with Attributes) contains 50 animals of 30,475 images. Following [45], we use the subset of 10,158 images from 50 classes with two types of 4096-d deep features extracted via DECAF and VGG19, respectively.

## E.2  Detailed Description of Baselines

- **SLIM** (Semi-supervised Multi-modal Learning with Incomplete Modalities) [46] simultaneously trains classifiers for different views and learns a consensus label matrix using the view-specific similarity matrices of existing instances.

- **AMSC** (Absent Multi-view Semi-supervised Classification) [19] learns view-specific label matrices and a shared label probability matrix incorporating intra-view and extra-view similarity losses with the $p$-th root integration strategy.

- **AMMSS** (Adaptive MultiModel Semi-Supervised classification) [7] is a GMvSSL method that simultaneously learns a consensus label matrix and view-specific weights using label propagation and the Laplacian graphs of each view.

- **AMGL** (Autoweighted Multiple Graph Learning) [8] automatically learns a set of weights for all the view-specific graphs without additional parameters and then integrates them into a consensus graph.

- **MLAN** (Multi-view Learning with Adaptive Neighbors) [9] simultaneously assigns weights for each view and learns the optimal local structure by modifying the view-specific similarity matrix during each iteration.

- **AMUSE** (Adaptive MUltiview SEmi-supervised model) [10] learns a fused graph through the view-specific graphs built previously and a prior structure without particular distribution assumption on the view weights.

- **FMSSL** (Fast Multi-view Semi-Supervised Learning) [11] learns a fused graph by using the similarity matrices of each view constructed through an anchor-based strategy.

- **FMSEL** (Flexible Multi-view SEmi-supervised Learning) [12] utilizes a linear penalty term to adaptively assign weights across views, effectively learning a well-structured fused graph.

- **CFMSC** (adaptive Collaborative Fusion for Multi-view Semi-supervised Classification) [13] simultaneously integrates multiple feature projections and similarity matrices in a collaborative fusion scheme to enhance the discriminative of the learned projection subspace, facilitating label propagation on the fused graph.

---

[2] https://www.vision.caltech.edu/datasets/cub_200_2011/

[3] https://archive.ics.uci.edu/dataset/72/multiple+features

[4] https://data.caltech.edu/records/mzrjq-6wc02

[5] https://figshare.com/articles/dataset/15-Scene_Image_Dataset/7007177

[6] https://cvml.ista.ac.at/AwA/

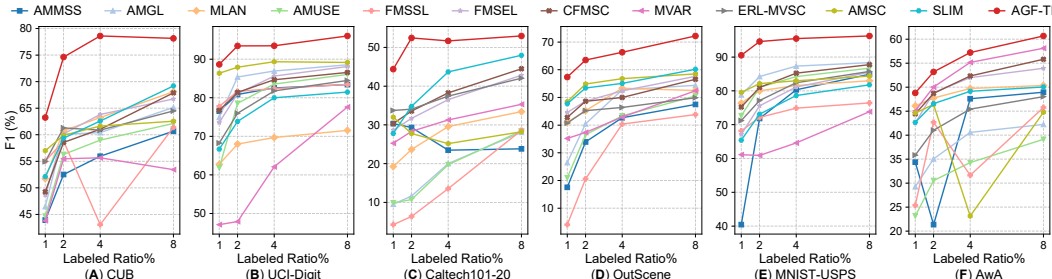

Figure 6: F1-score results on six datasets with LAR in $\{1\%, 2\%, 4\%, 8\%\}$ when VMR is 60%.

- **MVAR** (Multi-View semi-supervised classification via Adaptive Regression) [47] classifies the multi-view data by using the regression-based loss functions with $\ell_{2,1}$ norm.

- **ERL-MVSC** (Embedding Regularizer Learning scheme for Multi-View Semi-supervised Classification) [48] utilizes a linear regression model to obtain view-specific embedding regularizer and adaptively assigns view weights. The method then simultaneously learns a fused embedding regularizer by imposing $\ell_{2,1}$ norm and a shared label matrix for classification.

**Parameter setting for baselines.** For the above comparative methods, a grid search strategy is adopted to select the optimal parameters within the recommended range, and the recommended network structures are used as their baselines. Specifically, the structure of the deep neural network of DMF is set as $\sum_v^V d_v$-10*$c$-$c$, and the parameters $\beta$ and $\lambda$ are fixed to 0.01. For SLIM, we search optimal parameters $\lambda_1$ and $\lambda_2$ in the range of $\{10^{-4}, 10^{-3}, \ldots, 10^3, 10^4\}$. For AMSC, the parameter $p$ is fixed to 0.5 and $\gamma$ is searched from 1.1 to 3.1 with an interval of 0.4. For AMMSS, we search the logarithm of parameter $r$ from 0.1 to 2 with 0.4 step length and the regularization parameter $\lambda$ from 0 to 1 with an interval of 0.2. In MLAN, we randomly initialize the parameter $\lambda$ for the Laplacian matrix rank constraint to a positive value in the $[1, 30]$ interval. For AMUSE, the parameter $\lambda$ is tuned in $\{10^{-4}, 10^{-3}, \ldots, 10^3, 10^4\}$. In FMSSL, the parameter $\alpha$ is searched in $\{10^{-4}, 10^{-3}, \ldots, 10^2\}$, and the number of anchors $m$ is set to 1,024 when the sample size is larger than 10,000; $m$=256, otherwise. In FMSEL, the three parameters $\lambda_1$, $\lambda_2$, and $\xi$ are tuned in $\{10^{-6}, 10^{-2}, 10^0, 10^2, 10^6\}$. For CFMSC, we search the three parameters $\lambda$, $\beta$, $\gamma$ in $\{10^{-3}, 10^{-2}, \ldots, 10^2, 10^3\}$. For MVAR, we fix the parameter $r$ to 2 and the weight of unlabeled samples to $10^2$, while searching for the weight of labeled samples $\mu$ in $\{10^0, 10^1, \ldots, 10^6\}$. In ERL-MVSC, the smoothing factor $\alpha$, embedding parameter $\beta$, regularization parameter $\gamma$, and fitting coefficient $\delta$ are set to 2, 1, 1, and 10, respectively.

**Compute resources.** All the experimental environments are implemented on a desktop computer with two Intel Xeon Platinum 8488C CPUs and 256GB RAM, MATLAB 2022a (64-bit).

# F    Additional Results

## F.1    Main Results

In the manuscript, we evaluate the proposed method from the perspectives of view missing and label scarcity. For the view missing aspect, Tables 12, 13, and 14 present the detailed results corresponding to Table 2, including standard deviations. In addition, we perform the Wilcoxon rank-sum test at a 0.05 significance level to assess statistical reliability. The Wilcoxon rank-sum test results also validate the robustness of the proposed method in tackling the incomplete multi-view data.

We show the comparison results for the label scarcity setting in terms of F1 and Precision in Figs. 6 and 7. As observed, our method outperforms all other comparison methods over all metrics and all datasets, demonstrating the effectiveness of the proposed approach.

## F.2    Ablation Study

We present the detailed ablation results corresponding to Table 3, including standard deviations, on the six datasets in Tables 6, 7, and 8 in terms of Accuracy, F1 and Precision, respectively.

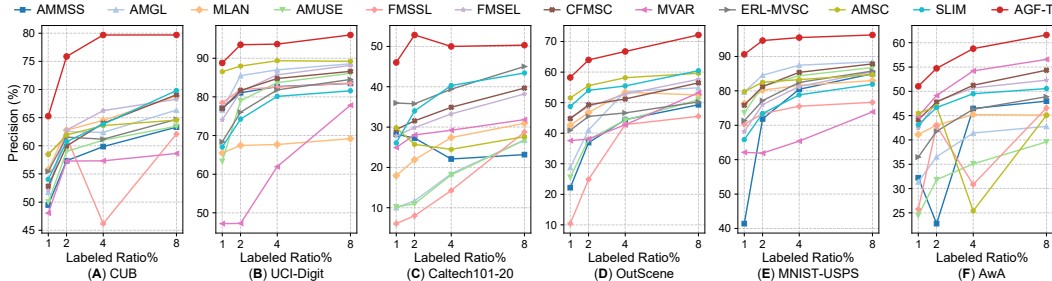

Figure 7: Precision results on six datasets with LAR in $\{1\%, 2\%, 4\%, 8\%\}$ when VMR is 60%.

Table 6: Accuracy ablation results (mean±std) of **AGF-TI** with different VMRs when LAR% is 5%.

| VMR=30% | CUB | UCI-Digit | Caltech101-20 | OutScene | MNIST-USPS | AwA | Avg. |
|---|---|---|---|---|---|---|---|
| **AGF-TI** | **78.33±3.46** | **95.98±0.82** | **81.88±1.26** | **70.34±1.86** | **95.09±0.79** | 69.12±0.81 | **81.79** |
| w/o $\mathbf{T}_v$ | 72.12±5.00 | 88.48±4.79 | 42.71±9.69 | 27.35±7.56 | 82.01±7.31 | **72.34±1.26** | 64.17 |
| w/o $\alpha_v$ | 76.77±1.95 | 90.28±6.32 | 69.65±25.73 | 66.41±1.45 | 93.24±0.81 | 66.49±1.15 | 77.14 |
| w/o TI | 72.82±2.56 | 88.96±2.35 | 76.63±0.82 | 63.55±1.16 | 88.85±0.70 | 59.23±1.22 | 75.01 |
| VMR=70% | | | | | | | |
| **AGF-TI** | **74.25±4.44** | **95.16±1.13** | **72.17±1.63** | **64.52±2.94** | **95.62±1.35** | 69.41±1.26 | **78.52** |
| w/o $\mathbf{T}_v$ | 53.40±14.97 | 55.58±16.99 | 38.54±3.65 | 30.62±4.31 | 78.12±11.94 | **70.58±1.09** | 54.47 |
| w/o $\alpha_v$ | 67.67±11.96 | 76.47±4.77 | 55.07±2.07 | 58.10±2.84 | 94.40±1.09 | 42.08±0.89 | 65.63 |
| w/o TI | 63.16±1.58 | 83.04±2.52 | 47.55±3.62 | 34.69±2.07 | 84.07±1.22 | 49.04±1.40 | 60.26 |

## F.3 Parameter Sensitivity Analysis

**Sensitivity analysis of the regularized parameters $\beta_\lambda$ and $\rho$.** We show the parameter sensitivity results on the six datasets under F1, and Precision metrics in Figs. 8 and 9.These results also support the statement discussed in Section 4.4 that $\beta_\lambda$ exhibits more sensitive effects on the classification performance compared to $\rho$. Besides, the proposed approach consistently achieves stable and competitive performance within a narrow parameter range across diverse datasets, demonstrating the

Table 7: F1-score ablation results (mean±std) of **AGF-TI** with different VMRs when LAR% is 5%.

| VMR=30% | CUB | UCI-Digit | Caltech101-20 | OutScene | MNIST-USPS | AwA | Avg. |
|---|---|---|---|---|---|---|---|
| **AGF-TI** | **77.03±4.39** | **95.97±0.82** | **58.02±3.91** | **69.56±2.53** | **95.07±0.80** | 57.16±1.26 | **75.47** |
| w/o $\mathbf{T}_v$ | 70.61±6.09 | 88.34±4.93 | 21.61±7.95 | 26.91±7.70 | 81.70±7.49 | **59.81±1.67** | 58.16 |
| w/o $\alpha_v$ | 76.05±2.29 | 90.35±6.09 | 48.13±23.59 | 66.59±1.42 | 93.14±0.85 | 56.09±1.34 | 71.72 |
| w/o TI | 71.87±3.00 | 88.94±2.36 | 50.56±2.65 | 63.06±1.57 | 88.76±0.70 | 50.59±1.73 | 68.96 |
| VMR=70% | | | | | | | |
| **AGF-TI** | **72.30±5.62** | **95.13±1.16** | **39.72±3.71** | **63.29±3.01** | **95.59±1.37** | 57.59±1.38 | **70.60** |
| w/o $\mathbf{T}_v$ | 51.19±15.30 | 54.56±16.90 | 8.29±1.87 | 27.11±4.35 | 77.61±12.38 | 56.83±1.37 | 45.93 |
| w/o $\alpha_v$ | 64.84±13.70 | 75.59±5.90 | 22.86±2.17 | 57.41±2.90 | 94.33±1.14 | 35.02±1.04 | 58.34 |
| w/o TI | 61.97±1.72 | 82.97±2.55 | 17.90±3.62 | 33.02±2.56 | 83.92±1.23 | 46.99±1.31 | 54.46 |

Table 8: Precision ablation results (mean±std) of **AGF-TI** with different VMRs when LAR% is 5%.

| VMR=30% | CUB | UCI-Digit | Caltech101-20 | OutScene | MNIST-USPS | AwA | Avg. |
|---|---|---|---|---|---|---|---|
| **AGF-TI** | **78.33±3.46** | **95.98±0.82** | **56.75±3.33** | **69.99±2.21** | **95.09±0.79** | 58.47±0.89 | **75.77** |
| w/o $\mathbf{T}_v$ | 72.12±5.00 | 88.48±4.79 | 21.35±8.43 | 27.41±7.85 | 82.01±7.31 | **61.43±1.44** | 58.80 |
| w/o $\alpha_v$ | 76.77±1.95 | 90.28±6.32 | 48.15±24.28 | 67.11±1.41 | 93.24±0.81 | 57.90±1.22 | 72.24 |
| w/o TI | 72.82±2.56 | 88.96±2.35 | 50.07±2.36 | 63.11±1.30 | 88.85±0.70 | 50.38±1.47 | 69.03 |
| VMR=70% | | | | | | | |
| **AGF-TI** | **74.25±4.44** | **95.16±1.13** | **36.81±3.04** | **63.83±2.89** | **95.62±1.35** | 58.82±1.33 | **70.75** |
| w/o $\mathbf{T}_v$ | 53.40±14.97 | 55.58±16.99 | 8.86±1.62 | 29.05±4.47 | 78.12±11.94 | **59.05±1.10** | 47.34 |
| w/o $\alpha_v$ | 67.67±11.96 | 76.47±4.77 | 20.97±1.97 | 58.09±2.79 | 94.40±1.09 | 36.54±0.84 | 59.02 |
| w/o TI | 63.16±1.58 | 83.04±2.52 | 15.86±3.05 | 32.37±2.14 | 84.07±1.22 | 42.61±1.34 | 53.52 |

generalizability of our method. This observation provides a useful guideline for parameter selection, enhancing its practicality in real-world applications.

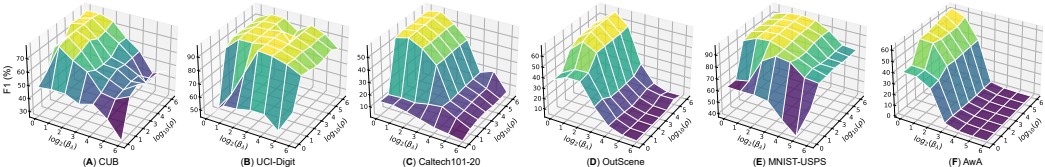

Figure 8: Parameter sensitivity analysis of $\beta_\lambda$ and $\rho$ in terms of F1.

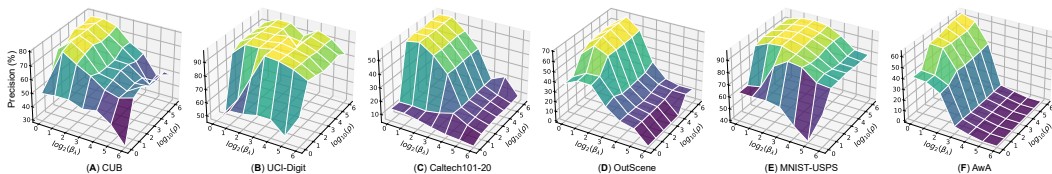

Figure 9: Parameter sensitivity analysis of $\beta_\lambda$ and $\rho$ in terms of Precision.

**Sensitivity analysis of the anchor number** $m$. To examine the influence of $m$, we vary its value in the range of $\{2^4, 2^5, 2^6, 2^7, 2^8\}$ for CUB, and $\{2^6, 2^7, 2^8, 2^9, 2^{10}\}$ for the other five datasets. The results for Accuracy, F1, and Precision metrics are shown in Figs. 10, 11, and 12. We observe a consistent trend where performance initially improves with increasing $m$, followed by a gradual decline, which aligns with the behavior of $\beta_\lambda$. For lower $m$, the small anchor-based bipartite graphs in each view are too coarse to capture the fine-grained geometric structures of the existing data points, limiting classification performance. Conversely, a large $m$ leads to sparser graph connectivity due to a fixed number of neighbors $k$, resulting in unstable information propagation. These results suggest that an appropriately selected $m$ balances local structural preservation with adequate graph connectivity.

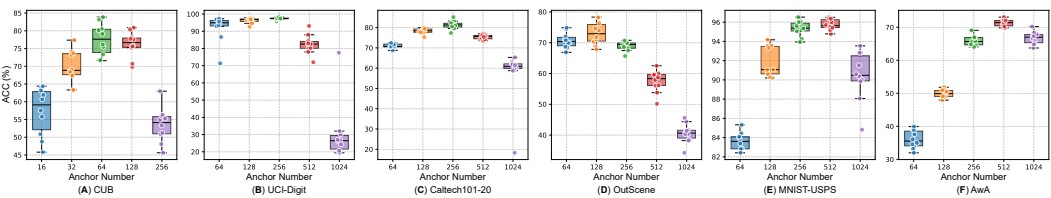

Figure 10: ACC results of **AGF-TI** with different anchor numbers.

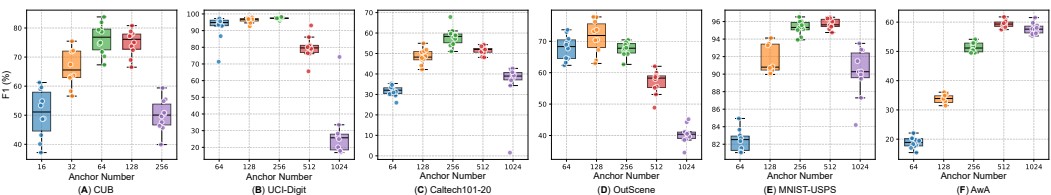

Figure 11: F1 results of **AGF-TI** with different anchor numbers.

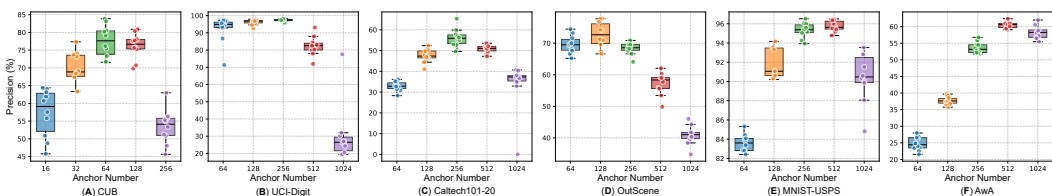

Figure 12: Precision results of **AGF-TI** with different anchor numbers.

**Sensitivity analysis of the trade-off parameter** $\lambda$. We investigate the impact of $\lambda$ by tuning its value within the set $\{V^2/3, V^2/2, V^2, 2V^2, 3V^2\}$, where $V$ is the number of views. The results for Accuracy, F1 and Precision on the six datasets are displayed in Figs. 13, 14, and 15, respectively. It is evident that the trade-off parameter $\lambda$ substantially affects performance, highlighting the crucial role of the graph fusion term, *i.e.*, AGF. In addition, we find that our method consistently achieves the highest performance across all datasets under different metrics when $\lambda$ is $V^2$. Based on this finding, we simply fix $\lambda$ to $V^2$ in our experiments.

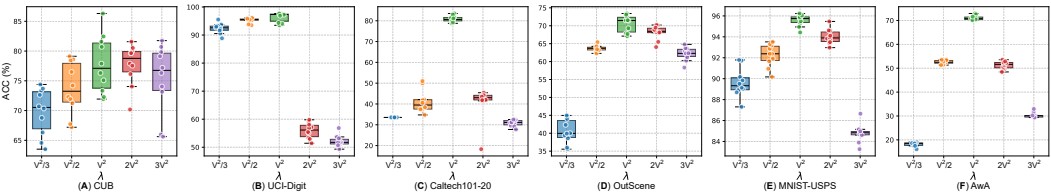

Figure 13: Parameter sensitivity analysis of $\lambda$ in terms of Accuracy.

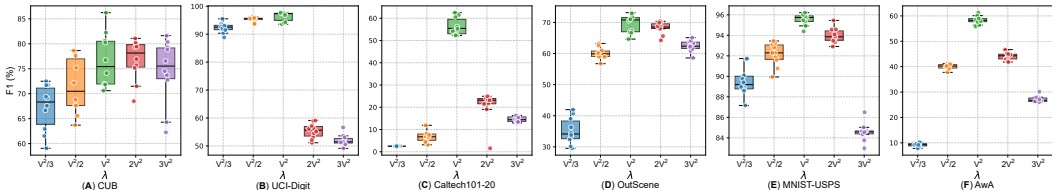

Figure 14: Parameter sensitivity analysis of $\lambda$ in terms of F1.

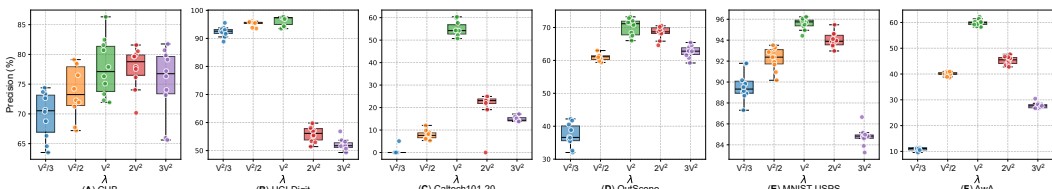

Figure 15: Parameter sensitivity analysis of $\lambda$ in terms of Precision.

### F.4 Empirical Analysis of *Sufficient Inner Optimization* Assumption

In Appendix D.1, we prove that the variable sequence obtained by Algorithm 2 converges to a stationary point based on the assumption 1, *i.e.*, Sufficient Inner Optimization. To validate the reasonable of this assumption, we track the error of $\boldsymbol{\alpha}$ and $\mathbf{P}$ between iteration steps, *i.e.*, $\|\boldsymbol{\alpha}^{(k+1)} - \boldsymbol{\alpha}^{(k)}\|_2^2$ and $\|\mathbf{P}^{(k+1)} - \mathbf{P}^{(k)}\|_F^2$ during Algorithm 1 to approximate the error between numerical and exact solutions in Eq. (36). The results for all datasets with VMR = 50% and LAR = 5% are shown in Table 9. One could observe that the error of both $\boldsymbol{\alpha}$ and $\mathbf{P}$ can be rapidly decreased to a small number (*e.g.*, 1e-5), suggesting the reasonableness of the sufficient optimization assumption.

### F.5 Computational and Memory Footprint of the Tensor Nuclear-Norm Step

As analyzed in Appendix D.2, the complexity of the optimization phase is primarily determined by the tensor nuclear-norm (TNN) step for solving the $\mathcal{G}$-subproblem. To further validate the scalability of the proposed **AGF-TI**, we utilize the YTF50 dataset (~126k samples) [53] with four views and empirically analyze the computational and memory footprint of the TNN step as the number of views, anchors, and samples scale simultaneously. The results under VMR = 50% and LAR = 5% are shown in Table 10. As is evident, the empirical trend aligns well with our theoretical analysis. Due to the anchor strategy, the running time of the TNN step does not increase dramatically with the sample size and remains within an acceptable burden. This indicates **AGF-TI** is capable of practical use.

Table 9: Iterative error of $\alpha$ and $\mathbf{P}$ during Algorithm 1 under VMR=50% and LAR=5%. ('-' means converged.)

| Step | | 1 | 2 | 3 | 4 | 5 | 6 |
|------|------|------|------|------|------|------|------|
| **CUB** | $\alpha$ error | 1.02e-5 | - | - | - | - | - |
| | $\mathbf{P}$ error | 2.83e-10 | - | - | - | - | - |
| **UCI-Digit** | $\alpha$ error | 2.14e-6 | 4.34e-7 | 5.17e-8 | - | - | - |
| | $\mathbf{P}$ error | 6.82e-5 | 1.31e-12 | 2.02e-17 | - | - | - |
| **Caltech101-20** | $\alpha$ error | 1.36e-4 | 6.71e-5 | 3.26e-5 | 1.53e-5 | 7.61e-6 | 3.22e-6 |
| | $\mathbf{P}$ error | 0.87 | 7.31e-9 | 7.56e-15 | 1.62e-20 | 1.34e-25 | 3.12e-30 |
| **OutScene** | $\alpha$ error | 1.41e-4 | 4.67e-5 | 1.58e-5 | 5.20e-6 | 1.73e-6 | - |
| | $\mathbf{P}$ error | 1.88e-5 | 1.8e-10 | 3.48e-15 | 8.06e-20 | 2.02e-24 | - |
| **MNIST-USPS** | $\alpha$ error | 6.33e-6 | - | - | - | - | - |
| | $\mathbf{P}$ error | 1.41e-4 | - | - | - | - | - |
| **AwA** | $\alpha$ error | 0.016 | 8.52e-8 | 1.22e-7 | 8.52e-8 | 1.22e-7 | 8.52e-8 |
| | $\mathbf{P}$ error | 0.004 | 3.84e-4 | 4.50e-5 | 7.46e-6 | 1.10e-6 | 3.31e-7 |

Table 10: Computational and memory footprint of the tensor nuclear-norm step with varying numbers of views, anchors, and samples under VMR=50% and LAR=5%.

| | #Anchor | Time (s) | | | | Memory (GiB) | | | |
|------|------|------|------|------|------|------|------|------|------|
| | | 128 | 256 | 512 | 1,024 | 128 | 256 | 512 | 1,024 |
| | | | | **#Sample=10k** | | | | | |
| #View | 2 | 0.35 | 0.42 | 0.76 | 1.17 | 0.095 | 0.13 | 0.38 | 0.76 |
| | 3 | 0.35 | 0.49 | 0.88 | 1.57 | 0.14 | 0.29 | 0.57 | 1.15 |
| | 4 | 0.48 | 0.67 | 1.05 | 1.95 | 0.19 | 0.38 | 0.76 | 1.53 |
| | | | | **#Sample=70k** | | | | | |
| #View | 2 | 1.93 | 3.02 | 4.56 | 8.89 | 1.17 | 2.18 | 4.28 | 8.56 |
| | 3 | 2.11 | 3.67 | 6.38 | 14.74 | 1.61 | 3.21 | 6.42 | 12.87 |
| | 4 | 2.98 | 4.91 | 9.06 | 23.10 | 2.14 | 4.28 | 8.56 | 17.13 |
| | | | | **#Sample≈126k** | | | | | |
| #View | 2 | 1.73 | 5.75 | 9.40 | 19.30 | 1.93 | 3.89 | 7.71 | 15.48 |
| | 3 | 2.33 | 7.28 | 16.23 | 33.00 | 3.02 | 6.03 | 11.59 | 23.26 |
| | 4 | 3.07 | 9.20 | 21.29 | 60.44 | 3.85 | 7.71 | 15.42 | 30.86 |

### F.6 Additional Comparisons to Deep Learning Baselines

We conduct additional experiments with two deep learning-based multi-view semi-supervised methods, *i.e.*, IMvGCN [54] and GEGCN [55], with VMR=50% and LAR=5%. We adopt the recommended learning rate and network structures as their baselines. Table 11 shows that our AGF-TI also outperforms them in almost all cases, further demonstrating the effectiveness of AGF-TI.

Table 11: Comparisons of the two deep learning baselines under VMR=50% and LAR=5%.

| Method | CUB | UCI-Digit | Caltech101-20 | OutScene | MNIST-USPS | AwA |
|--------|-----|-----------|---------------|----------|------------|-----|
| **Accuracy** | | | | | | |
| IMvGCN | 63.3 | 82.7 | 68.6 | 53.7 | 79.9 | 65.8 |
| GEGCN | 49.9 | 90.1 | 67.1 | 55.3 | 91.3 | 58.8 |
| **Ours** | **80.2** | **95.2** | **80.0** | **69.2** | **95.6** | **70.6** |
| **Precision** | | | | | | |
| IMvGCN | 67.7 | 83.5 | 51.2 | 53.7 | 81.1 | **60.8** |
| GEGCN | 56.5 | 90.6 | 34.0 | 56.2 | 91.4 | 57.0 |
| **Ours** | **80.2** | **95.2** | **53.1** | **68.3** | **95.6** | 60.0 |
| **F1-score** | | | | | | |
| IMvGCN | 61.2 | 82.5 | 40.2 | 52.4 | 79.5 | **58.6** |
| GEGCN | 50.5 | 90.1 | 31.6 | 55.5 | 91.3 | 50.4 |
| **Ours** | **79.1** | **95.3** | **54.8** | **67.7** | **95.6** | **58.6** |

Table 12: Comparison results (mean±std) of compared methods on CUB and UCI-Digit under different VMRs when fix LAR to 5%.

| Method | CUB | | | | | | | | | UCI-Digit | | | | | | | | |
|---|---|---|---|---|---|---|---|---|---|---|---|---|---|---|---|---|---|---|
| | VMR=30% | | | VMR=50% | | | VMR=70% | | | | | | | | | | | |
| | ACC | PREC | F1 | ACC | PREC | F1 | ACC | PREC | F1 | | | | | | | | | |
| AMMSS | 67.33±1.82 ● | 67.33±1.82 ● | 64.57±1.66 ● | 48.44±16.67 ● | 48.44±16.67 ● | 44.25±18.35 ● | 62.74±4.03 ● | 62.74±4.03 ● | 60.92±5.09 ● | | | | | | | | | |
| AMGL | 67.11±2.79 ● | 67.11±2.79 ● | 64.64±3.47 ● | 65.77±4.15 ● | 65.77±4.15 ● | 63.32±4.91 ● | 61.75±2.12 ● | 61.75±2.12 ● | 60.04±2.89 ● | | | | | | | | | |
| MLAN | 70.09±3.30 ● | 70.09±3.30 ● | 68.35±4.11 ● | 67.75±2.29 ● | 67.75±2.29 ● | 66.04±3.71 ● | 62.25±3.99 ● | 62.25±3.99 ● | 60.68±4.53 ● | | | | | | | | | |
| AMUSE | 64.26±3.68 ● | 64.26±3.68 ● | 61.84±4.60 ● | 63.21±3.85 ● | 63.21±3.85 ● | 61.35±4.69 ● | 62.77±2.49 ● | 62.77±2.49 ● | 60.90±3.58 ● | | | | | | | | | |
| FMSSL | 65.30±4.09 ● | 65.30±4.09 ● | 62.31±5.92 ● | 48.72±25.57 ● | 45.72±30.12 ● | 44.62±28.36 ● | 64.49±2.34 ● | 64.49±2.34 ● | 62.68±2.90 ● | | | | | | | | | |
| FMSEL | 70.54±2.73 ● | 70.54±2.73 ● | 69.21±3.16 ● | 67.28±2.86 ● | 67.28±2.86 ● | 66.16±3.33 ● | 62.35±3.12 ● | 62.35±3.12 ● | 60.81±3.38 ● | | | | | | | | | |
| CFMSC | 71.65±3.16 ● | 71.65±3.16 ● | 70.81±3.33 ● | 69.19±3.70 ● | 69.19±3.70 ● | 68.02±4.21 ● | 55.23±15.26 ● | 54.23±18.23 ● | 53.13±17.35 ● | | | | | | | | | |
| MVAR | 66.49±2.90 ● | 66.49±2.90 ● | 65.19±3.23 ● | 61.51±3.09 ● | 61.51±3.09 ● | 60.20±3.15 ● | 50.18±5.07 ● | 50.18±5.07 ● | 41.15±7.40 ● | | | | | | | | | |
| ERL-MVSC | 65.75±1.86 ● | 65.75±1.86 ● | 65.48±1.86 ● | 61.51±3.26 ● | 61.51±3.26 ● | 61.34±3.35 ● | 59.98±2.33 ● | 59.98±2.33 ● | 59.73±2.49 ● | | | | | | | | | |
| AMSC | 68.75±3.22 ● | 68.75±3.22 ● | 66.14±4.69 ● | 67.60±3.47 ● | 67.60±3.47 ● | 65.95±4.52 ● | 64.39±2.15 ● | 64.39±2.15 ● | 63.08±2.72 ● | | | | | | | | | |
| SLIM | 68.30±2.98 ● | 68.30±2.98 ● | 67.36±3.34 ● | 65.12±2.62 ● | 65.12±2.62 ● | 63.48±3.36 ● | 64.04±2.65 ● | 64.04±2.65 ● | 63.42±2.73 ● | | | | | | | | | |
| **Ours** | **78.33±3.46** | **78.33±3.46** | **77.03±4.39** | **80.23±4.77** | **80.23±4.77** | **79.10±5.49** | **74.25±4.44** | **74.25±4.44** | **72.30±5.62** | | | | | | | | | |
| AMMSS | 91.46±0.96 ● | 91.46±0.96 ● | 91.43±0.98 ● | 87.30±0.72 ● | 87.30±0.72 ● | 87.24±0.72 ● | 76.04±3.04 ● | 76.04±3.04 ● | 75.87±2.95 ● | | | | | | | | | |
| AMGL | 91.06±0.98 ● | 91.06±0.98 ● | 90.93±1.07 ● | 89.73±0.82 ● | 89.73±0.82 ● | 89.65±0.86 ● | 85.32±0.70 ● | 85.32±0.70 ● | 85.22±0.72 ● | | | | | | | | | |
| MLAN | 86.13±3.46 ● | 86.13±3.46 ● | 86.27±3.50 ● | 74.82±5.76 ● | 74.82±5.76 ● | 76.31±4.68 ● | 62.43±2.14 ● | 62.43±2.14 ● | 65.09±1.91 ● | | | | | | | | | |
| AMUSE | 90.61±1.11 ● | 90.61±1.11 ● | 90.50±1.18 ● | 87.21±1.30 ● | 87.21±1.30 ● | 87.13±1.32 ● | 81.91±1.34 ● | 81.91±1.34 ● | 81.81±1.35 ● | | | | | | | | | |
| FMSSL | 91.12±0.70 ● | 91.12±0.70 ● | 91.07±0.73 ● | 86.72±0.74 ● | 86.72±0.74 ● | 86.62±0.76 ● | 78.00±2.42 ● | 78.00±2.42 ● | 77.72±2.54 ● | | | | | | | | | |
| FMSEL | 92.19±1.42 ● | 92.19±1.42 ● | 92.16±1.41 ● | 88.99±0.98 ● | 88.99±0.98 ● | 88.95±0.99 ● | 83.53±1.41 ● | 83.53±1.41 ● | 83.40±1.40 ● | | | | | | | | | |
| CFMSC | 91.48±1.37 ● | 91.48±1.37 ● | 91.44±1.36 ● | 88.14±1.18 ● | 88.14±1.18 ● | 88.08±1.19 ● | 83.47±1.16 ● | 83.47±1.16 ● | 83.33±1.14 ● | | | | | | | | | |
| MVAR | 80.51±1.69 ● | 80.51±1.69 ● | 80.43±1.66 ● | 73.46±1.82 ● | 73.46±1.82 ● | 73.11±1.85 ● | 68.01±3.56 ● | 68.01±3.56 ● | 67.78±3.59 ● | | | | | | | | | |
| ERL-MVSC | 86.80±1.23 ● | 86.80±1.23 ● | 86.78±1.24 ● | 84.61±1.31 ● | 84.61±1.31 ● | 84.59±1.29 ● | 79.20±1.43 ● | 79.20±1.43 ● | 79.20±1.43 ● | | | | | | | | | |
| AMSC | 93.87±0.60 ● | 93.87±0.60 ● | 93.86±0.60 ● | 91.21±0.50 ● | 91.21±0.50 ● | 91.19±0.51 ● | 87.59±1.01 ● | 87.59±1.01 ● | 87.55±1.01 ● | | | | | | | | | |
| SLIM | 84.93±0.79 ● | 84.93±0.79 ● | 84.80±0.85 ● | 81.41±1.04 ● | 81.41±1.04 ● | 81.42±1.02 ● | 72.55±1.24 ● | 72.55±1.24 ● | 72.43±1.26 ● | | | | | | | | | |
| **Ours** | **95.98±0.82** | **95.98±0.82** | **95.97±0.82** | **95.23±2.99** | **95.23±2.99** | **95.25±2.94** | **95.16±1.13** | **95.16±1.13** | **95.13±1.16** | | | | | | | | | |
| Win ●/Tie ⊙/Loss ○ | 66/0/0 | | | 66/0/0 | | | 66/0/0 | | | | | | | | | | | |

1 ●/⊙/○ denote that the proposed method performs significantly better/tied/worse than the baselines by the one-sided Wilcoxon rank-sum test with confidence level 0.05.

Table 13: Comparison results (mean±std) of compared methods on Caltech101-20 and OutScene under different VMRs when fix LAR to 5%.

**Caltech101-20**

| Method | VMR=30% | | | VMR=50% | | | VMR=70% | | |
|---|---|---|---|---|---|---|---|---|---|
| | ACC | PREC | F1 | ACC | PREC | F1 | ACC | PREC | F1 |
| AMMSS | 69.20±0.99 ● | 30.41±1.70 ● | 32.19±2.63 ● | 65.87±1.02 ● | 25.59±1.58 ● | 26.30±2.08 ● | 60.87±1.68 ● | 20.39±2.42 ● | 20.57±2.78 ● |
| AMGL | 61.85±1.12 ● | 22.88±1.40 ● | 24.97±2.01 ● | 62.18±1.73 ● | 22.50±1.67 ● | 24.06±2.01 ● | 61.60±1.22 ● | 21.51±1.50 ● | 22.56±2.08 ● |
| MLAN | 74.07±0.75 ● | 39.74±1.52 ● | 43.49±2.24 ● | 69.92±0.67 ● | 33.26±1.27 ● | 36.07±1.94 ● | 63.60±1.27 ● | 25.42±1.84 ● | 26.95±2.01 ● |
| AMUSE | 60.52±1.28 ● | 21.87±0.97 ● | 24.22±1.34 ● | 61.85±1.61 ● | 22.27±2.18 ● | 23.95±2.72 ● | 60.42±0.76 ● | 20.55±1.15 ● | 21.40±1.89 ● |
| FMSSL | 69.28±2.58 ● | 30.60±4.52 ● | 30.36±4.53 ● | 43.20±14.95 ● | 10.51±7.36 ● | 9.87±7.14 ● | 57.01±1.56 ● | 15.71±1.16 ● | 15.11±0.99 ● |
| FMSEL | 77.86±0.69 ● | 47.16±1.64 ● | 51.59±1.89 ● | 72.63±0.67 ● | 37.77±1.36 ● | 41.09±1.64 ● | 66.92±0.42 ● | 29.27±0.98 ● | 31.49±1.20 ● |
| CFMSC | 78.89±0.67 ● | 49.39±1.67 ● | 53.70±2.03 ● | 73.46±0.74 ● | 39.55±1.58 ● | 43.15±2.03 ● | 67.76±0.68 ● | 31.11±1.45 ● | 33.90±1.95 ● |
| MVAR | 78.81±0.89 ● | 49.27±2.09 ● | 53.60±2.44 ● | 72.08±0.97 ● | 36.78±2.11 ● | 40.08±2.45 ● | 63.92±0.79 ● | 24.57±0.95 ● | 25.17±1.56 ● |
| ERL-MVSC | 65.81±1.62 ● | 49.78±2.99 ● | 46.96±2.64 ● | 62.69±1.24 ● | 44.69±2.03 ● | 42.23±1.96 ● | 55.47±1.14 ● | 34.95±1.51 ⊙ | 33.60±1.56 ● |
| AMSC | 63.73±3.12 ● | 22.51±3.19 ● | 22.60±3.19 ● | 65.04±0.73 ● | 24.34±0.92 ● | 23.53±1.50 ● | 64.79±0.66 ● | 26.02±0.95 ● | 26.52±1.50 ● |
| SLIM | 76.58±1.13 ● | 48.71±1.78 ● | 52.94±2.07 ● | 73.02±0.70 ● | 44.50±1.55 ● | 48.48±1.55 ● | 65.17±0.90 ● | 34.86±1.62 ● | 37.39±2.10 ⊙ |
| **Ours** | **81.88±1.26** | **56.75±3.33** | **58.02±3.91** | **79.99±1.63** | **53.10±3.41** | **54.79±3.69** | **72.17±1.63** | **36.81±3.04** | **39.72±3.71** |
| Win ●/Tie ⊙/Loss ○ | | 66/0/0 | | | 66/0/0 | | | 64/2/0 | |

**OutScene**

| Method | VMR=30% | | | VMR=50% | | | VMR=70% | | |
|---|---|---|---|---|---|---|---|---|---|
| | ACC | PREC | F1 | ACC | PREC | F1 | ACC | PREC | F1 |
| AMMSS | 58.11±1.71 ● | 55.68±1.78 ● | 53.77±1.87 ● | 53.47±1.15 ● | 51.16±1.26 ● | 49.54±1.51 ● | 44.43±2.04 ● | 42.16±2.03 ● | 40.56±2.67 ● |
| AMGL | 58.28±2.32 ● | 57.71±2.58 ● | 58.32±2.48 ● | 56.19±1.93 ● | 55.58±2.22 ● | 55.54±2.54 ● | 48.33±2.66 ● | 47.84±2.92 ● | 46.69±3.11 ● |
| MLAN | 64.08±1.07 ● | 63.89±1.23 ● | 63.85±1.42 ● | 57.11±1.60 ● | 57.00±1.71 ● | 56.81±1.74 ● | 43.57±3.52 ● | 42.90±3.99 ● | 42.04±4.53 ● |
| AMUSE | 56.80±1.93 ● | 56.11±1.92 ● | 57.04±1.77 ● | 55.17±2.14 ● | 54.64±2.24 ● | 54.80±1.95 ● | 48.70±2.02 ● | 48.43±2.20 ● | 47.54±2.03 ● |
| FMSSL | 52.88±8.00 ● | 51.34±8.25 ● | 50.16±10.20 ● | 48.73±3.24 ● | 47.52±3.85 ● | 46.61±4.19 ● | 42.76±1.77 ● | 40.95±2.02 ● | 38.90±2.46 ● |
| FMSEL | 66.10±1.11 ● | 66.03±1.02 ● | 66.10±1.14 ● | 56.42±2.36 ● | 56.17±2.57 ● | 56.46±2.05 ● | 50.34±1.66 ● | 49.54±1.73 ● | 49.14±1.41 ● |
| CFMSC | 67.41±1.45 ● | 67.07±1.53 ● | 66.84±1.59 ● | 58.84±1.47 ● | 58.15±1.56 ● | 57.96±1.76 ● | 47.59±2.50 ● | 46.89±2.49 ● | 45.67±1.93 ● |
| MVAR | 62.91±1.36 ● | 62.85±1.46 ● | 62.34±1.60 ● | 58.42±1.46 ● | 58.39±1.58 ● | 57.36±1.56 ● | 51.61±1.41 ● | 51.16±1.52 ● | 49.73±1.83 ● |
| ERL-MVSC | 52.99±0.97 ● | 53.60±0.96 ● | 53.27±0.92 ● | 48.91±1.24 ● | 49.34±1.24 ● | 49.04±1.25 ● | 42.58±1.36 ● | 43.07±1.42 ● | 42.72±1.41 ● |
| AMSC | 65.15±1.25 ● | 63.95±1.39 ● | 62.48±1.97 ● | 60.33±0.77 ● | 59.54±0.89 ● | 58.84±1.03 ● | 52.65±0.94 ● | 51.93±0.96 ● | 50.93±1.11 ● |
| SLIM | 66.75±1.48 ● | 65.92±1.64 ● | 65.16±2.22 ● | 61.65±1.20 ● | 60.98±1.23 ● | 60.67±1.27 ● | 52.69±0.51 ● | 52.53±0.60 ● | 52.22±0.71 ● |
| **Ours** | **70.34±1.86** | **69.99±2.21** | **69.56±2.53** | **69.24±2.65** | **68.29±2.89** | **67.70±3.47** | **64.52±2.94** | **63.83±2.89** | **63.29±3.01** |
| Win ●/Tie ⊙/Loss ○ | | 66/0/0 | | | 66/0/0 | | | 64/0/0 | |

[1] ●/⊙/○ denote that the proposed method performs significantly better/tied/worse than the baselines by the one-sided Wilcoxon rank-sum test with confidence level 0.05.

Table 14: Comparison results (mean±std) of compared methods on MNIST-USPS and AwA under different VMRs when fix LAR to 5%.

**MNIST-USPS**

| Method | VMR=30% | | | VMR=50% | | | VMR=70% | | |
|---|---|---|---|---|---|---|---|---|---|
| | ACC | PREC | F1 | ACC | PREC | F1 | ACC | PREC | F1 |
| AMMSS | 90.20±0.61 ● | 90.20±0.61 ● | 90.13±0.61 ● | 84.93±1.17 ● | 84.93±1.17 ● | 84.80±1.20 ● | 81.43±1.79 ● | 81.43±1.79 ● | 81.29±1.77 ● |
| AMGL | 93.54±0.60 ● | 93.54±0.60 ● | 93.50±0.61 ● | 89.61±0.74 ● | 89.61±0.74 ● | 89.53±0.75 ● | 85.75±0.32 ● | 85.75±0.32 ● | 85.61±0.32 ● |
| MLAN | 88.29±1.18 ● | 88.29±1.18 ● | 88.24±1.23 ● | 84.96±1.98 ● | 84.96±1.98 ● | 84.88±2.05 ● | 75.72±4.16 ● | 75.72±4.16 ● | 75.92±3.84 ● |
| AMUSE | 93.38±0.37 ● | 93.38±0.37 ● | 93.35±0.38 ● | 88.39±0.79 ● | 88.39±0.79 ● | 88.28±0.80 ● | 83.16±0.77 ● | 83.16±0.77 ● | 82.99±0.82 ● |
| FMSSL | 86.36±0.43 ● | 86.36±0.43 ● | 86.17±0.46 ● | 79.72±1.76 ● | 79.72±1.76 ● | 79.15±1.86 ● | 72.58±2.12 ● | 72.58±2.12 ● | 71.76±2.26 ● |
| FMSEL | 90.51±0.59 ● | 90.51±0.59 ● | 90.47±0.60 ● | 85.40±0.97 ● | 85.40±0.97 ● | 85.32±0.98 ● | 80.39±0.91 ● | 80.39±0.91 ● | 80.22±0.91 ● |
| CFMSC | 93.55±0.52 ● | 93.55±0.52 ● | 93.53±0.52 ● | 89.05±0.51 ● | 89.05±0.51 ● | 88.97±0.54 ● | 84.57±0.93 ● | 84.57±0.93 ● | 84.45±0.91 ● |
| MVAR | 79.65±2.10 ● | 79.65±2.10 ● | 79.54±2.08 ● | 71.73±1.78 ● | 71.73±1.78 ● | 71.40±2.00 ● | 66.04±3.82 ● | 66.04±3.82 ● | 66.08±3.87 ● |
| ERL-MVSC | 88.02±0.62 ● | 88.02±0.62 ● | 88.00±0.62 ● | 85.90±0.96 ● | 85.90±0.96 ● | 85.85±0.96 ● | 82.46±0.49 ● | 82.46±0.49 ● | 82.41±0.52 ● |
| AMSC | 88.72±0.87 ● | 88.72±0.87 ● | 88.56±0.91 ● | 83.16±3.76 ● | 83.16±3.76 ● | 83.00±3.53 ● | 83.83±1.06 ● | 83.83±1.06 ● | 83.60±1.11 ● |
| SLIM | 84.61±1.02 ● | 84.61±1.02 ● | 84.45±1.06 ● | 81.33±0.77 ● | 81.33±0.77 ● | 81.17±0.77 ● | 79.17±0.78 ● | 79.17±0.78 ● | 78.93±0.78 ● |
| **Ours** | **95.09±0.79** | **95.09±0.79** | **95.07±0.80** | **95.58±0.82** | **95.58±0.82** | **95.55±0.84** | **95.62±1.35** | **95.62±1.35** | **95.59±1.37** |

**AwA**

| Method | VMR=30% | | | VMR=50% | | | VMR=70% | | |
|---|---|---|---|---|---|---|---|---|---|
| | ACC | PREC | F1 | ACC | PREC | F1 | ACC | PREC | F1 |
| AMMSS | 63.48±0.81 ● | 53.67±0.88 ● | 53.86±1.14 ● | 58.52±0.54 ● | 49.16±0.70 ● | 49.73±1.03 ● | 55.21±0.70 ● | 46.33±0.74 ● | 47.22±0.87 ● |
| AMGL | 57.74±0.85 ● | 49.25±0.81 ● | 48.78±0.97 ● | 52.94±0.81 ● | 44.82±0.92 ● | 44.07±1.13 ● | 47.45±1.87 ● | 40.09±1.55 ● | 39.47±1.61 ● |
| MLAN | 66.49±0.55 ● | 58.22±0.77 ⊙ | 59.25±0.99 ○ | 56.74±0.89 ● | 49.44±0.73 ● | 52.78±0.70 ● | 47.72±0.66 ● | 41.82±0.54 ● | 48.37±0.68 ● |
| AMUSE | 51.44±1.93 ● | 44.02±1.75 ● | 44.06±1.62 ● | 46.50±1.58 ● | 39.31±1.29 ● | 38.79±1.19 ● | 40.51±3.69 ● | 34.31±3.18 ● | 33.51±3.19 ● |
| FMSSL | 65.60±0.44 ● | 55.79±0.68 ● | 55.32±0.95 ● | 59.34±0.39 ● | 50.30±0.49 ● | 50.20±0.77 ● | 52.24±0.63 ● | 42.86±0.74 ● | 41.57±0.85 ● |
| FMSEL | 65.32±0.63 ● | 58.86±0.54 ⊙ | 60.09±0.56 ○ | 58.33±0.64 ● | 52.04±0.71 ● | 53.53±0.79 ● | 57.03±2.28 ● | 50.71±2.03 ● | 51.72±1.69 ● |
| CFMSC | 67.24±0.47 ● | 59.88±0.51 ○ | 60.92±0.61 ○ | 62.23±0.58 ● | 54.71±0.61 ● | 55.69±0.79 ● | 59.85±0.66 ● | 52.41±0.76 ● | 53.31±0.92 ● |
| MVAR | 69.07±0.55 ⊙ | **61.87±0.85** ● | **62.91±1.08** ● | 63.57±0.61 ● | 56.99±0.61 ● | 58.23±0.78 ⊙ | 59.15±2.45 ● | 52.35±2.37 ● | 54.92±0.57 ● |
| ERL-MVSC | 55.02±0.73 ● | 51.33±0.98 ● | 50.51±0.84 ● | 52.12±0.89 ● | 48.43±0.82 ● | 47.63±0.83 ● | 49.11±0.66 ● | 45.32±0.65 ● | 44.54±0.63 ● |
| AMSC | 64.34±0.39 ● | 54.74±0.57 ● | 54.06±0.86 ● | 55.90±0.73 ● | 45.76±0.68 ● | 44.83±0.85 ● | 52.94±0.52 ● | 43.29±0.67 ● | 42.52±0.95 ● |
| SLIM | 64.78±0.36 ● | 54.24±0.54 ● | 53.50±0.80 ● | 61.40±0.69 ● | 51.28±0.76 ● | 50.73±0.75 ● | 59.01±0.26 ● | 49.35±0.35 ● | 48.81±0.32 ● |
| **Ours** | **69.12±0.81** | 58.47±0.89 | 57.16±1.26 | **70.58±1.15** | **59.77±1.50** | **58.62±1.67** | **69.41±1.26** | **58.82±1.33** | **57.59±1.38** |
| Win ●/Tie ⊙/Loss ○ | 57/3/6 | | | 65/1/0 | | | 66/0/0 | | |

[1] ●/⊙/○ denote that the proposed method performs significantly better/tied/worse than the baselines by the one-sided Wilcoxon rank-sum test with confidence level 0.05.