# OpenReview forum: "Adversarial Graph Fusion for Incomplete Multi-view Semi-supervised Learning with Tensorial Imputation"
_NeurIPS.cc/2025/Conference — NeurIPS 2025 poster_

### Official Review · Reviewer_dtsk · 2025-06-25

**Clarity:** 2
**Significance:** 2
**Originality:** 2
**Rating:** 4
**Confidence:** 4

**Summary:**

This paper tackles the challenge of missing views in multi-view semi-supervised learning, which often disrupts local data structures and impairs classification accuracy. The authors introduce AGF-TI, a novel framework that combats structural discontinuities by generating a unified graph through adversarial learning. It leverages high-order consistency via tensor-based structure recovery and incorporates anchor points to enhance efficiency. Extensive experiments confirm that AGF-TI consistently outperforms existing methods across multiple benchmarks.

**Questions:**

Please refer to the weaknesses.

**Ethical Concerns:**

["NO or VERY MINOR ethics concerns only"]

**Final Justification:**

The authors basically addressed my concerns, so I decide to raise my score.

**Limitations:**

yes

**Quality:**

3

**Strengths And Weaknesses:**

Strengths:
1. The writing and presentation are good, then the readers can quickly grasp the core idea of the paper.
2. The experiments are sufficient to verify the effectiveness of the proposed method.

Weaknesses:
1. The anchor learning and low-rank tensor decomposition have been widely studies in multi-view clustering, I cannot see some insightful observations and interesting idea.
2. The authors demonstrate the Sub-Cluster Problem on a Synthetic dataset. However, this could be artificially crafted. Hence, the such verification should be conducted on real-world datasets.
3. The most used multi-view datasets are too small, it’s suggested to add more large-scale datasets.
4. The author claims that introducing anchors can decrease the computational burden, then the running time should be compared.

---

> ### Author Rebuttal · Authors · 2025-07-30
>
> Thank you so much for the valuable comments! We are committed to addressing each question you have raised.
>
> ---
>
> ### **W1: Novelty of the Proposed AGF-TI**
>
> We acknowledge that both anchor learning and low-rank tensor decomposition are existing techniques in the multi-view clustering field. Actually, in this paper, the anchor strategy and tensor method are utilized to enhance efficiency and unify the framework. Beyond combining these techniques, our work mainly focuses on introducing a novel **adversarial graph fusion** (AGF) operator based on anchors to learn a robust consensus graph for semi-supervised classification, specifically designed to address the sub-cluster problem. The proposed AGF offers the following advantages:
>
> - **Anchor graph fusion paradigm**
>
> Most existing multi-view semi-supervised learning (MvSSL) methods [10, 12] directly fuse the sample-level graphs $\mathbf{S}_v\in \mathbb{R}^{n \times n}$. In contrast, AGF-TI operates on anchor-based bipartite graphs of size $n \times m$, enabling fusion at a lower dimensional level. Besides, the subsequent tensor learning can also be conducted on an $m\times V\times n$ tensor instead of $n\times V\times n$. This paradigm significantly enhances the computational efficiency for both graph fusion and tensor imputation.
>
> - **Adversarial min-max framework**
>
> We are the first to introduce a min-max optimization framework into graph-based MvSSL, enabling the learning of a robust consensus graph. This novel approach mitigates the impact of distorted local structures and improves the robustness of graph fusion. Unlike conventional min-min paradigms, our adversarial min-max cycle avoids collapse into a single dominant view and achieves a better *exploration–exploitation* balance, ultimately leading to improved classification performance.
>
> In addition to the proposed AGF, we also present a severe yet neglected problem, the Sub-Cluster Problem, when encountering view missing scenarios.
>
> **Sub-cluster observation.** We identify a novel challenge in graph-based MvSSL with missing views, called Sub-Cluster Problem (SCP). Specifically, missing samples in individual views can violate the core smoothness assumption underlying label propagation, resulting in inaccurate graph fusion and poor label prediction. Moreover, we validate the presence of the SCP in real-world webpage and document classification datasets in the next **W2**.
>
> **Other contributions.** We develop an effective optimization algorithm that combines alternating direction method of multipliers and reduced gradient descent method to tackle the challenges posed by the proposed objective function. Furthermore, we provide a theoretical convergence guarantee under a mild assumption, highlighting the soundness of our approach.
>
> ---
>
> ### **W2: SCP in Real-world Datasets**
>
> Thanks. To address the reviewer’s concern, we validate the SCP using two real-world datasets: WebKB [R1] and Wikipedia [R2], commonly used in webpage and document classification tasks. Specifically, WebKB comprises 1,051 webpages across 2 classes, 230 for course and 821 for no-course, with page and link views. Wikipedia contains 693 multimedia documents from 10 categories, represented by image and text views.
>
> To quantify the number of category clusters in each view, we employ the density-based clustering algorithm DBSCAN [R3]. For each view, we calibrated DBSCAN at VMR=0% by setting $minpts=5$ and tuning $\epsilon \in [0.1, 1]$ (step size 0.01) to approximate the real number of classes. To account for the reduction in sample size at higher VMRs, we decrement $minpts$ by one for each increase in VMR. The results are shown below, and one can observe that the number of detected clusters by DBSCAN rapidly grows with VMR, indicating that class manifolds fragment into smaller sub-clusters. Notably, in the ''link'' view of WebKB, DBSCAN identifies 11 clusters even at VMR=0% (more than 2 classes), which suggests that SCP may not only be widely present in incomplete multi-view data but also arise in complete data scenarios. These findings support the validity and generality of SCP in real-world applications.
>
> | WebKB         | #class | 0%   | 30%  | 50%  | 70%  |
> | ------------- | ------ | ---- | ---- | ---- | ---- |
> | page view     | 2      | 2    | 3    | 4    | 9    |
> | link view     | 2      | 11   | 15   | 19   | 27   |
> | **Wikipedia** |        |      |      |      |      |
> | image view    | 10     | 7    | 9    | 11   | 16   |
> | text view     | 10     | 10   | 11   | 15   | 22   |
>
> ---
>
> ### **W3: Additional Large-scale Datasets**
>
> Thanks. Actually, the datasets in our experiments are commonly used in the multi-view learning field, exhibiting considerable variation in sample sizes (600 to 10,158). To address the reviewer's concern, we conduct additional experiments on the NoisyMNIST datasets (30k/70k samples, 2 views, 10 classes) [R4] to evaluate the classification performance of compared algorithms on larger-scale datasets. With VMR and LAR set to 50% and 5%, respectively, the results below demonstrate that AGF-TI continues to achieve superior performance, confirming its scalability and effectiveness.
>
> | NoisyMNIST |       | 30K   |       |       | 70K   |       |
> | ---------- | ----- | ----- | ----- | ----- | ----- | ----- |
> | **Method**     | **ACC**   | **PREC**  | **F1**    | **ACC**   | **PREC**  | **F1**    |
> | AMMSS      | 60.72 | 59.74 | 62.59 | -     | -     | -     |
> | AMGL       | 51.8  | 51.02 | 52.11 | -     | -     | -     |
> | MLAN       | -     | -     | -     | -     | -     | -     |
> | AMUSE      | 60.29 | 59.63 | 59.5  | -     | -     | -     |
> | FMSSL      | 78.12 | 77.44 | 77.11 | -     | -     | -     |
> | FMSEL      | 52.66 | 51.94 | 51.51 | -     | -     | -     |
> | CFMSC      | 72.67 | 72.11 | 71.67 | -     | -     | -     |
> | MVAR       | 61.81 | 61.15 | 60.42 | 68.21 | 67.39 | 67.35 |
> | ERL-MVSC   | 77.31 | 77.04 | 76.83 | -     | -     | -     |
> | AMSC       | 72.04 | 71.34 | 73.09 | 71.45 | 70.88 | 72.75 |
> | SLIM       | 66.74 | 66.18 | 65.63 | -     | -     | -     |
> | **Ours**       | **81.48** | **81.01** | **81.08** | **77.34** | **77.78** | **76.75** |
>
> Note: Some results were not reported due to out-of-memory errors or because the computations had not been completed within the allotted time.
>
> ---
>
>
> ### **W4: Running Time**
>
> Thanks. To address the reviewer's concern, we compared algorithm running times on all datasets, setting VMR=50% and LAR=5%. For baselines requiring complete multi-view data, the total running time includes both the DMF completion process and their subsequent execution. The results are recorded below, and one can observe that AGF-TI maintains an acceptable running time. Although AGF-TI takes slightly longer than AMSC and SLIM due to its missing data imputation and shared bipartite graph learning, it consistently achieves significantly better performance across all datasets. Moreover, due to its anchor strategy, AGF-TI's running time scales efficiently with the sample size, ensuring its practicality for large-scale applications.
>
>
>
> | Time (s) | CUB  | UCI-Digit | Caltech101-20 | OutScene | MNIST-USPS | AwA     |
> | -------- | ---- | --------- | ------------- | -------- | ---------- | ------- |
> | AMMSS    | 4.31 | 7.80      | 86.25         | 15.61    | 48.58      | 621.34  |
> | AMGL     | 4.16 | 5.51      | 65.99         | 9.60     | 39.89      | 587.51  |
> | MLAN     | 4.68 | 20.29     | 141.65        | 74.99    | 150.06     | 1068.73 |
> | AMUSE    | 6.37 | 31.77     | 102.76        | 55.59    | 209.93     | 1361.76 |
> | FMSSL    | 4.77 | 5.81      | 82.94         | 24.51    | 74.75      | 808.80  |
> | FMSEL    | 6.45 | 32.70     | 120.18        | 60.12    | 234.50     | 1359.60 |
> | CFMSC    | 6.78 | 20.36     | 106.95        | 26.29    | 159.53     | 815.56  |
> | MVAR     | 4.26 | 5.40      | 69.84         | 10.36    | 39.30      | 601.27  |
> | ERL-MVSC | 4.31 | 5.68      | 67.71         | 10.55    | 42.66      | 605.83  |
> | AMSC     | 0.85 | 2.15      | 11.03         | 4.40     | 3.73       | 33.54   |
> | SLIM     | 0.33 | 4.46      | 35.59         | 14.12    | 11.43      | 128.83  |
> | **Ours** | 1.06 | 15.35     | 45.03         | 35.84    | 31.06      | 354.54  |
>
>
> **References**
>
>
>
> [R1] Blum, Avrim, and Tom Mitchell. "Combining labeled and unlabeled data with co-training." Proc. Conf. Learn. Theory, 1998.
>
> [R2] Wang, Hao, et al. "Multi-view clustering via concept factorization with local manifold regularization." IEEE ICDM, 2016.
>
> [R3] Ester, Martin, et al. "A density-based algorithm for discovering clusters in large spatial databases with noise." kdd, 1996.
>
> [R4] Yang, Mouxing, et al. "Robust multi-view clustering with incomplete information." IEEE TPAMI, 2022.

---

> > ### Comment · Reviewer_dtsk · 2025-08-04
> > **Response**
> >
> > I appreaciate many efforts made by the authors. Although I sitll think the novelty is limieted, I'm glad to raise my score.

---

> ### Author Response · Authors · 2025-08-04
> **Thank you!**
>
> Thank you for taking the time to review our rebuttal and raising the score. We sincerely appreciate your support and endorsement!

---

### Official Review · Reviewer_Z8gs · 2025-06-27

**Clarity:** 3
**Significance:** 3
**Originality:** 3
**Rating:** 5
**Confidence:** 4

**Summary:**

This paper proposes AGF-TI, an innovative approach for incomplete multi-view semi-supervised learning, to address the Sub-Cluster Problem (SCP) caused by missing views. By integrating adversarial graph fusion (min-max optimization) and tensorial imputation (low-rank tensor learning), AGF-TI reconstructs robust consensus graphs and recovers missing structures. Combined with an anchor-based strategy for computational efficiency, AGF-TI demonstrates superior performance on various datasets, validating its effectiveness in handling both view incompleteness and label scarcity challenges.

**Questions:**

See the weaknesses.

**Ethical Concerns:**

["NO or VERY MINOR ethics concerns only"]

**Final Justification:**

The authors' response have addressed most of my concerns, I will maintain my score as 5.

**Limitations:**

yes

**Quality:**

3

**Strengths And Weaknesses:**

Strengths:
1. This paper identifies the Sub-Cluster Problem in graph-based multi-view semi-supervised learning, an important but previously under-explored challenge.
2. AGF-TI innovatively combines min-max scheme with tensorial imputation, forming a robust framework to address SCP.
3. Clear and mathematically rigorous formulation of the optimization problem, with a solid convergence analysis in the appendix.
4. Experimental results demonstrate a substantial improvement in classification performance. The paper also includes a detailed ablation study to demonstrate the effectiveness of each component in AGF-TI.

Weaknesses:
1. Although the time complexity is discussed, it would be better to provide the running time results of the AGF-TI and baselines for comparison.
2. What does the parameter $\beta$ mean in Eq. (3)?
3. How are the permutation matrices $\mathbf{T}_v$ initialized in practice?

---

> ### Author Rebuttal · Authors · 2025-07-30
>
> Thank you so much for your valuable comments! We are committed to addressing each question you have raised.
>
> ---
>
> ### **W1: Running Time**
>
> Thanks. To address the reviewer's concern, we compared algorithm running times on all datasets, setting VMR=50% and LAR=5%. For baselines requiring complete multi-view data, the total running time includes both the DMF completion process and their subsequent execution. The results are recorded below, and it can be observed that AGF-TI maintains an acceptable running time. Although AGF-TI takes slightly longer than AMSC and SLIM due to its missing data imputation and shared bipartite graph learning, it consistently achieves significantly better performance across all datasets. Moreover, due to its anchor strategy, AGF-TI's running time scales efficiently with the sample size, ensuring its practicality for large-scale applications.
>
> | Time (s) | CUB  | UCI-Digit | Caltech101-20 | OutScene | MNIST-USPS | AwA     |
> | -------- | ---- | --------- | ------------- | -------- | ---------- | ------- |
> | AMMSS    | 4.31 | 7.80      | 86.25         | 15.61    | 48.58      | 621.34  |
> | AMGL     | 4.16 | 5.51      | 65.99         | 9.60     | 39.89      | 587.51  |
> | MLAN     | 4.68 | 20.29     | 141.65        | 74.99    | 150.06     | 1068.73 |
> | AMUSE    | 6.37 | 31.77     | 102.76        | 55.59    | 209.93     | 1361.76 |
> | FMSSL    | 4.77 | 5.81      | 82.94         | 24.51    | 74.75      | 808.80  |
> | FMSEL    | 6.45 | 32.70     | 120.18        | 60.12    | 234.50     | 1359.60 |
> | CFMSC    | 6.78 | 20.36     | 106.95        | 26.29    | 159.53     | 815.56  |
> | MVAR     | 4.26 | 5.40      | 69.84         | 10.36    | 39.30      | 601.27  |
> | ERL-MVSC | 4.31 | 5.68      | 67.71         | 10.55    | 42.66      | 605.83  |
> | AMSC     | 0.85 | 2.15      | 11.03         | 4.40     | 3.73       | 33.54   |
> | SLIM     | 0.33 | 4.46      | 35.59         | 14.12    | 11.43      | 128.83  |
> | **Ours** | 1.06 | 15.35     | 45.03         | 35.84    | 31.06      | 354.54  |
>
>
> ---
>
>
>
> ### **W2: $\beta$ in Eq. (3)**
>
> Thanks for pointing it out. In Eq. (3), $\beta$ is the trade-off parameter of the regularization term $||\mathbf{P}||_ F^2$. In our final objective, it is multiplied by $\lambda$ and written as $\beta_\lambda$. We will clarify this in the revision.
>
> ---
>
>
>
> ### **W3: Initialization of $\mathbf{T}_v$**
>
> Thanks. In our experiments, the permutation matrix $\mathbf{T}_v$ is initialized as an identity matrix.
>
> **Last**
>
> Thank you again for your suggestion. We will include these details in the revised paper.

---

> ### Comment · Reviewer_Z8gs · 2025-08-04
> **Replying to Authors**
>
> The authors' response have addressed most of my concerns, I will maintain my score as 5.

---

> ### Author Response · Authors · 2025-08-04
> **Thank you!**
>
> Thank you for taking the time to review our rebuttal. We're glad our clarifications addressed your concerns, and we sincerely appreciate your support and endorsement! We will carefully revise the paper to incorporate the discussed points.

---

### Official Review · Reviewer_MMoo · 2025-06-29

**Clarity:** 3
**Significance:** 3
**Originality:** 3
**Rating:** 5
**Confidence:** 4

**Summary:**

The paper proposes Adversarial Graph Fusion with Tensorial Imputation (AGF-TI) for Incomplete Multi-view Semi-supervised Learning. This paper addresses a newly identified issue termed Sub-Cluster Problem (SCP). SCP arises when missing views in multi-view data disrupt the smoothness assumption of label propagation by creating disconnected sub-clusters. AGF-TI addresses SCP through i) an adversarial graph fusion scheme leveraging a min-max optimization framework to fuse view-specific graphs and 2) a tensorial imputation mechanism that recovers missing similarity information using high-order correlations. It also employ acceleration strategy to reduce the computational cost.

**Questions:**

If the authors could address the above weaknesses during rebuttal, I would like to keep my rating.

**Ethical Concerns:**

["NO or VERY MINOR ethics concerns only"]

**Final Justification:**

The responses have addressed my concerns, I will increase my score.

**Limitations:**

yes

**Quality:**

3

**Strengths And Weaknesses:**

Strength:

1. Extensive experiments on six public datasets under various missing-view and label-scarcity settings show that AGF-TI consistently outperforms state-of-the-art methods in terms of accuracy, precision, and F1-score.
2. The paper offers a comprehensive optimization algorithm combining reduced gradient descent and ADMM, supported by theoretical convergence analysis.

Weaknesses

1. The experimental protocol creates missing views by randomly selecting V% of the samples and randomly deleting 1 to V – 1 of their views. In Figure 1, however, the Sub-Cluster Problem is illustrated as arising when whole groups of neighboring samples are missing, leaving a "vacuum region" that fractures a class manifold into multiple sub-clusters and breaks the smoothness assumption of label propagation. If missing views are indeed independent and spatially random, contiguous voids, thus SCP should occur far less frequently. Could the authors quantify how often SCP actually appears under the current random-missing setup, or ii) provide experiments with more realistic, block-missing / structured-missing patterns to show that AGF-TI remains effective in scenarios where SCP is genuinely likely to occur?
2. What is the computational and memory footprint of the tensor nuclear-norm step when the number of views, anchors, and samples all scale simultaneously? Specifically, can the authors validate them on a truly large-scale (>100 k samples) benchmark?
3. The related-work section should discuss recent missing-view representation methods, such as "COMPLETER: Incomplete Multi-view Clustering via Contrastive Prediction" and "Decoupled Contrastive Multi-view Clustering with High-order Random Walks."
4. It would be better to give the full name of TNN in Definition 1.
5. The evaluation lacks deep learning baselines. Benchmarking only against classical or shallow methods limits the paper’s relevance to modern multi-view semi-supervised learning.
6. In Definition 1, please spell out TNN’s full name Tensor Nuclear Norm.

---

> ### Author Rebuttal · Authors · 2025-07-27
>
> Thank you for your acknowledgment and insightful comments! Your feedback is extremely helpful, and we are committed to addressing each question you have raised.
>
> ---
>
> ### **W1: Show SCP Appears under Current Random-missing Setup**
>
> Thanks. Our experimental setup replicates that of previous work [17] to ensure a fair comparison. To address the reviewer's concern, we adopt a density-based adaptive clustering method [R1], DBSCAN, to validate that "current random-missing setup effectively mimics SCP as VMR increases".
>
> For validation, we calibrated DBSCAN on the first view of each dataset at VMR=0% by setting $minpts=5$ and tuning $\epsilon \in [0.1, 1]$ (step size 0.01) to approximate the real number of classes. To control for the sample size reduction accompanying higher VMRs, we decrement $minpts$ by one for each increase in VMR. The results are shown below, and one can observe that the number of detected clusters by DBSCAN rapidly grows with VMR, indicating that class manifolds fragment into smaller sub-clusters. This confirms our experimental setup can effectively simulate SCP and validates the effectiveness of our proposed AGF-TI for addressing SCP.
>
> | Dataset       | #Classes | 0%   | 30%  | 50%  | 70%  |
> | ------------- | :-------: | ---- | ---- | ---- | ---- |
> | CUB           | 10       | 9    | 9    | 13   | 17   |
> | UCI-Digit     | 10       | 10   | 11   | 18   | 32   |
> | Caltech101-20 | 20       | 20   | 25   | 31   | 47   |
> | OutScene      | 8        | 8    | 7    | 10   | 18   |
> | MNIST-USPS    | 10       | 10   | 11   | 13   | 27   |
> | AwA           | 50       | 43   | 46   | 62   | 126  |
>
>
>
> ---
>
>
>
> ### **W2: Computational and Memory Footprint of the Tensor Nuclear-Norm Step**
>
> Thanks. Theoretically, we have discussed the time complexity of the tensor nuclear-norm (TNN) step in Appendix D.2, which is $\mathcal{O}(nmV\text{log}(nV)+nmV^2)$. To address the reviewer's concern, we adopt the YTF50 dataset (~126k) [R2] with four views and empirically analyze the computational and memory footprint of the TNN step when view number, anchor number, and sample size scale simultaneously. The results under VMR=50% and LAR=5% are shown below. As is evident, the empirical trend aligns well with our theoretical analysis. Due to the anchor strategy, the running time of TNN step does not increase dramatically with the sample size and remains within an acceptable burden. This indicates AGF-TI is capable for practical use.
>
> - **#Sample = 10k**
>
> | Time (s)       |      |      |      |       | Memory (GiB)   |       |      |      |       |
> | :--------------: | ---- | ---- | ---- | ----- | :--------------: | ----- | ---- | ---- | ----- |
> | **#view\\#anchor** | **128**  | **256**  | **512**  | **1,024** | **#view\\#anchor** | **128**   | **256**  | **512**  | **1,024** |
> | 2              | 0.35 | 0.42 | 0.76 | 1.17  | 2              | 0.095 | 0.13 | 0.38 | 0.76  |
> | 3              | 0.35 | 0.49 | 0.88 | 1.57  | 3              | 0.14  | 0.29 | 0.57 | 1.15  |
> | 4              | 0.48 | 0.67 | 1.05 | 1.95  | 4              | 0.19  | 0.38 | 0.76 | 1.53  |
>
> - **#Sample = 70k**
>
> | Time (s)       |      |      |      |       | Memory (GiB)   |      |      |      |       |
> | :--------------: | ---- | ---- | ---- | ----- | :--------------: | ----- | ---- | ---- | ----- |
> | **#view\\#anchor** | **128**  | **256**  | **512**  | **1,024** | **#view\\#anchor** | **128**   | **256**  | **512**  | **1,024** |
> | 2              | 1.93 | 3.02 | 4.56 | 8.89  | 2              | 1.17 | 2.18 | 4.28 | 8.56  |
> | 3              | 2.11 | 3.67 | 6.38 | 14.74 | 3              | 1.61 | 3.21 | 6.42 | 12.87 |
> | 4              | 2.98 | 4.91 | 9.06 | 23.10  | 4              | 2.14 | 4.28 | 8.56 | 17.13 |
>
> - **#Sample = 126k**
>
> | Time (s)       |      |      |       |       | Memory (GiB)   |      |      |       |       |
> | :--------------: | ---- | ---- | ---- | ----- | :--------------: | ----- | ---- | ---- | ----- |
> | **#view\\#anchor** | **128**  | **256**  | **512**  | **1,024** | **#view\\#anchor** | **128**   | **256**  | **512**  | **1,024** |
> | 2              | 1.73 | 5.75 | 9.40   | 19.30  | 2              | 1.93 | 3.89 | 7.71  | 15.48 |
> | 3              | 2.33 | 7.28 | 16.23 | 33.00  | 3              | 3.02 | 6.03 | 11.59 | 23.26 |
> | 4              | 3.07 | 9.20  | 21.29 | 60.44 | 4              | 3.85 | 7.71 | 15.42 | 30.86 |
>
>
>
> ---
>
> ### **W3: References**
>
> Thanks for pointing them out. We will cite and discuss these papers in the revision.
>
> ---
>
>
>
>
> ### **W4/6: Full Name of TNN in Definition1**
>
> Thanks for pointing it out. We will revise TNN to its full name in Definition 1.
>
> ---
>
>
>
>
> ### **W5: Additional Deep Learning Baselines**
>
> Thanks. To address the reviewer's concern, we conduct additional experiments with two deep learning-based multi-view semi-supervised methods, i.e., IMvGCN [R3] and GEGCN [R4], on all datasets with VMR=50% and LAR=5%. We adopt the recommended learning rate and network structures as their baselines. The table below shows that our AGF-TI outperforms them in almost all cases, further demonstrating the effectiveness of AGF-TI.
>
> | ACC      | CUB  | UCI-Digit | Caltech101-20 | OutScene | MNIST-USPS | AwA  |
> | -------- | ---- | --------- | ------------- | -------- | ---------- | ---- |
> | IMvGCN   | 63.3 | 82.7      | 68.6          | 53.7     | 79.9       | 65.8 |
> | GEGCN    | 49.9 | 90.1      | 67.1          | 55.3     | 91.3       | 58.8 |
> | **Ours**     | **80.2** | **95.2**      | **80.0**            | **69.2**     | **95.6**       | **70.6** |
> | **PREC** |      |           |               |          |            |      |
> | IMvGCN   | 67.7 | 83.5      | 51.2          | 53.7     | 81.1       | **60.8** |
> | GEGCN    | 56.5 | 90.6      | 34.0            | 56.2     | 91.4       | 57.0   |
> | **Ours**     | **80.2** | **95.2**      | **53.1**          | **68.3**     | **95.6**       | 60.0   |
> | **F1**   |      |           |               |          |            |      |
> | IMvGCN   | 61.2 | 82.5      | 40.2          | 52.4     | 79.5       | 58.6 |
> | GEGCN    | 50.5 | 90.1      | 31.6          | 55.5     | 91.3       | 50.4 |
> | **Ours**     | **79.1** | **95.3**      | **54.8**          | **67.7**     | **95.6**       | **58.6** |
>
>
>
> **References**
>
> [R1] Ester, Martin, et al. "A density-based algorithm for discovering clusters in large spatial databases with noise." kdd, 1996.
>
> [R2] Huang, Dong, et al. "Fast multi-view clustering via ensembles: Towards scalability, superiority, and simplicity." IEEE TKDE, 2023.
>
> [R3] Wu, Zhihao, et al. "Interpretable graph convolutional network for multi-view semi-supervised learning." IEEE TMM, 2023.
>
> [R4] Lu, Jielong, et al. "Generative essential graph convolutional network for multi-view semi-supervised classification." IEEE TMM, 2024.

---

> > ### Comment · Reviewer_MMoo · 2025-08-04
> >
> > The responses have addressed my concerns, I will increase my score.

---

> ### Author Response · Authors · 2025-08-04
> **Thank you!**
>
> Thank you for taking the time to review our rebuttal and raising the score. We're glad our clarifications addressed your concerns, and we sincerely appreciate your support and endorsement! We will carefully revise the paper to incorporate the discussed points.

---

### Official Review · Reviewer_wnph · 2025-07-02

**Clarity:** 3
**Significance:** 3
**Originality:** 3
**Rating:** 4
**Confidence:** 4

**Summary:**

This paper introduces a novel method, Adversarial Graph Fusion with Tensorial Imputation (AGF-TI), to address the Sub-Cluster Problem (SCP) in graph-based multi-view semi-supervised learning (GMvSSL) where missing data can distort local structures and degrade classification performance. AGF-TI utilizes an adversarial min-max framework to learn a robust consensus graph that is resilient to these distortions. Simultaneously, it recovers missing structural information by stacking similarity graphs into a tensor and leveraging high-order correlations for imputation. The authors conduct extensive experiments to validate the effectiveness of AGF-TI in handling scenarios with both missing views and limited labeled data.

**Questions:**

1.The central formulation is a max-min-max problem. Could the authors elaborate on the intuition behind this structure? Specifically, how does the outer max over the view weights $\alpha_v$ in conjunction with the inner min over the fused graph $P$ lead to a more robust consensus graph than a simpler min-min formulation, as seen in methods like CFMSC?

2.Please clarify how the permutation matrix $T_v$ is initialized.

**Ethical Concerns:**

["NO or VERY MINOR ethics concerns only"]

**Final Justification:**

Thanks for the authors’ response. My concerns have been adequately addressed, and I decide to maintain my original score.

**Limitations:**

Yes

**Quality:**

4

**Strengths And Weaknesses:**

Strengths:

1.The paper is well-written, and the motivations are clearly described. The identification and formalization of a previously unaddressed challenge in this field, SCP, provide a strong motivation and a clear target for the proposed method.

2.The AGF-TI model is the first to adopt a min-max framework for graph fusion, which is a novel contribution to GMvSSL. Incorporates an anchor-based strategy to reduce the computational cost, making AGF-TI scalable to larger datasets.

3.I enjoyed the significance test in Tables 5, 6, and 7 in Appendix. The provided code and sufficient information make it easy to reproduce the experimental results.

Weaknesses:

1.The AGF-TI model is inherently complex, involving a tri-level max-min-max optimization problem.

2.The theoretical convergence relies on the sufficient optimization assumption.

---

> ### Author Rebuttal · Authors · 2025-07-30
>
> Thank you so much for the valuable comments! We are delighted that you found the strengths of AGF-TI. Below, we address each of your questions in detail.
>
> ---
>
> ### **W1/Q1: Max-Min-Max Framework and Its Benefits**
>
> Thanks. Compared to the traditional minimization paradigm, the proposed AGF-TI introduces an adversarial learning method to optimize the shared bipartite graph, thus formulated as a max-min-max framework. As discussed in Subsection 4.3, such adversarial learning enhances the performance and robustness of the approach. On the other hand, although our formulation looks intractable, it actually can be solved by an efficient optimization algorithm through ADMM and the reduced gradient descent method.
>
> In the traditional min-min paradigm, if a particular view temporarily achieves better performance, the algorithm tends to increase its weight, thereby diminishing the influence of other views (please see Fig. 3). This can lead to the algorithm becoming overly reliant on a single view, which may compromise its final performance.
>
> In contrast, our max-min-max framework creates an adversarial dynamic between maximizing the overall objective and minimizing the view weights $\boldsymbol{\alpha}$. This min-max cycle effectively prevents the algorithm from collapsing to a single dominant view, achieving *exploration–exploitation* balance. This mechanism enables the proposed AGF-TI to fully integrate complementary information from individual views to learn a stable and reliable fused bipartite graph and improve the final performance. Furthermore, as supported by recent studies [33, 36, 48], this adversarial property across views also enhances the model's robustness to noise when facing noisy views or datapoints.
>
> ---
>
>
>
> ### **W2: Sufficient Optimization Assumption**
>
> Thanks. According to Corollary 1 in Appendix D.1, Algorithm 1 will converge to the global optimum of the **Inner Step**, hence the sufficient optimization assumption is actually a mild assumption and can be quickly satisfied in practice. To validate the reasonable of this assumption, we track the error of both $\boldsymbol{\alpha}$ and $\mathbf{P}$ between iteration steps, i.e., $||\boldsymbol{\alpha}^{(k+1)}-\boldsymbol{\alpha}^{(k)}||_2^2$ and $||\mathbf{P}^{(k+1)}-\mathbf{P}^{(k)}||_F^2$ during Algorithm 1 to approximate the error between numerical and exact solutions in Eq. (36). The results on all datasets with VMR=50% and LAR=5% are shown below. These results show that the error of both $\boldsymbol{\alpha}$ and $\mathbf{P}$ can be rapidly decreased to a small number (e.g., $1e-5$), suggesting the reasonableness of the sufficient optimization assumption.
>
>
> | k                           | 1        | 2        | 3        | 4        | 5        | 6         |
> | --------------------------- | -------- | -------- | -------- | -------- | -------- | --------- |
> | **CUB**                     |          |          |          |          |          |           |
> | $\boldsymbol{\alpha}$ error | 1.02E-05 | \-       | \-       | \-       | \-       | \-        |
> | $\mathbf{P}$ error          | 2.83E-10 | \-       | \-       | \-       | \-       | \-        |
> | **UCI-Digit**               |          |          |          |          |          |           |
> | $\boldsymbol{\alpha}$ error | 2.14E-06 | 4.34E-07 | 5.17E-08 | \-       | \-       | \-        |
> | $\mathbf{P}$ error          | 6.82E-05 | 1.31E-12 | 2.02E-17 | \-       | \-       | \-        |
> | **Caltech101-20**           |          |          |          |          |          |           |
> | $\boldsymbol{\alpha}$ error | 1.36E-04 | 6.71E-05 | 3.26E-05 | 1.53E-05 | 7.61E-06 | 3.22E-06  |
> | $\mathbf{P}$ error          | 0.87     | 7.31E-09 | 7.56E-15 | 1.62E-20 | 1.34E-25 | 3.12e -30 |
> | **OutScene**                |          |          |          |          |          |           |
> | $\boldsymbol{\alpha}$ error | 1.41E-04 | 4.67E-05 | 1.58E-05 | 5.20E-06 | 1.73E-06 | \-        |
> | $\mathbf{P}$ error          | 1.88E-05 | 1.80E-10 | 3.48E-15 | 8.06E-20 | 2.02E-24 | \-        |
> | **MNIST-USPS**              |          |          |          |          |          |           |
> | $\boldsymbol{\alpha}$ error | 6.33E-06 | \-       | \-       | \-       | \-       | \-        |
> | $\mathbf{P}$ error          | 1.41E-04 | \-       | \-       | \-       | \-       | \-        |
> | **AwA**                     |          |          |          |          |          |           |
> | $\boldsymbol{\alpha}$ error | 0.016    | 8.52E-08 | 1.22E-07 | 8.52E-08 | 1.22E-07 | 8.52E-08  |
> | $\mathbf{P}$ error          | 0.004    | 3.84E-04 | 4.50E-05 | 7.46E-06 | 1.10E-06 | 3.31E-07  |
>
> Note: '-' means converged.
>
> ---
>
> ### **Q2: Initialization of $\mathbf{T}_v$**
>
> Thanks. In our experiments, the permutation matrix $\mathbf{T}_v$ is initialized as an identity matrix. We will include this detail in the revised paper.

---

> > ### Comment · Reviewer_wnph · 2025-08-09
> >
> > I thank the authors for their detailed responses and the additional experiments, which have well addressed my concerns regarding the intuition of max-min-max Framework and method details. Considering the contributions and quality of this work, I think it can make a significant contribution to the incomplete multi-view semi-supervised learning community. Therefore, I decide to maintain my original score.

---

> ### Author Response · Authors · 2025-08-09
> **Thank you!**
>
> Thank you for taking the time to review our paper and rebuttal. We're glad our clarifications addressed your concerns, and we sincerely appreciate your support and endorsement!

---

### Note · Authors · 2025-08-12

Dear AC and Reviewers,

We sincerely thank you for your thoughtful and constructive feedback. We have dedicated considerable time to crafting this rebuttal. Below, we summarize the key concerns raised and our corresponding responses:

- **Sufficient Optimization Assumption** (Reviewer `wnph`)
We provided an analysis of the numerical iteration error, demonstrating that this assumption is readily satisfied in practice.

- **Sub-Cluster Problem** (Reviewers `MMoo` and `dtsk`)
We demonstrated its severity and prevalence with experiments investigating the number of clusters, conducted across six original datasets and two new real-world datasets.

- **Additional Deep Learning Baselines** (Reviewer `MMoo`) **and Larger Datasets** (Reviewer `dtsk`)
We validated AGF-TI's effectiveness against two new deep learning baselines and on new larger-scale datasets (NoisyMNIST, up to 70k samples).

- **Running Time** (Reviewers `Z8gs` and `dtsk`) **and Scalability of TNN Step** (Reviewer `MMoo`)
We included running time and scalability analyses, confirming the efficiency of AGF-TI.

- **Implementation Details** (Reviewers `wnph` and `Z8gs`)
We clarified (1) the initialization of $\mathbf{T}_v$ and (2) the role of $\beta$ in Eq. (3).

We will include the revisions and additional experimental results in the revised paper.

Once again, we are grateful for your valuable insights, which have significantly enhanced our work. We have made every effort to comprehensively address all concerns.

Thank you for your time and consideration.

Sincerely,

The Authors

---

### Decision · Program_Chairs · 2025-09-17

**Decision:**

Accept (poster)

**Comment:**

After the rebuttal, all reviewers acknowledged that their concerns had been addressed and were consistently positive about this submission. I recommend accepting it. The authors are suggested to improve the camera-ready version according to the discussion.